# NON-REVERSIBLE PARALLEL TEMPERING FOR UNCERTAINTY APPROXIMATION IN DEEP LEARNING

## ABSTRACT

Parallel tempering (PT), also known as replica exchange, is the go-to workhorse for simulations of multi-modal distributions. The key to the success of PT is to adopt efficient swap schemes. The popular deterministic even-odd (DEO) scheme exploits the non-reversibility property and has successfully reduced the communication cost from $O(P^2)$ to $O(P)$ given sufficient many $P$ chains. However, such an innovation largely disappears in big data problems due to the limited chains and extremely few bias-corrected swaps. To handle this issue, we generalize the DEO scheme to *promote the non-reversibility* and obtain *an appealing communication cost $O(P \log P)$* based on the optimal window size. In addition, we also analyze the bias when we adopt stochastic gradient descent (SGD) with *large and constant* learning rates as exploration kernels. Such a *user-friendly nature* enables us to conduct large-scale uncertainty approximation tasks without much tuning costs.

## 1 INTRODUCTION

Langevin diffusion is a standard sampling algorithm that follows a stochastic differential equation

$$d\boldsymbol{\beta}_t = -\nabla U(\boldsymbol{\beta}_t)dt + \sqrt{2\tau}d\boldsymbol{W}_t,$$

where $\boldsymbol{\beta}_t \in \mathbb{R}^d$, $\nabla U(\cdot)$ is the gradient of the energy function $U(\cdot)$, $\boldsymbol{W}_t \in \mathbb{R}^d$ is a Brownian motion, and $\tau$ is the temperature. The diffusion process converges to a stationary distribution $\pi(\boldsymbol{\beta}) \propto e^{-\frac{U(\boldsymbol{\beta})}{\tau}}$ and setting $\tau = 1$ yields a Bayesian posterior. When $U(\cdot)$ is convex, the rapid convergence has been widely studied in Durmus & Moulines (2016); Dalalyan (2017); however, when $U(\cdot)$ is non-convex, a slow mixing rate is inevitable (Raginsky et al., 2017). To accelerate the simulation, replica exchange Langevin diffusion (reLD) proposes to include a high-temperature particle $\boldsymbol{\beta}_t^{(P)}$, where $P \in \mathbb{N}^+ \backslash \{1\}$, for *exploration*. Meanwhile, a low-temperature particle $\boldsymbol{\beta}_t^{(1)}$ is presented for *exploitation*:

$$\begin{aligned} d\boldsymbol{\beta}_t^{(P)} &= -\nabla U(\boldsymbol{\beta}_t^{(P)})dt + \sqrt{2\tau^{(P)}}d\boldsymbol{W}_t^{(P)}, \\ d\boldsymbol{\beta}_t^{(1)} &= -\nabla U(\boldsymbol{\beta}_t^{(1)})dt + \sqrt{2\tau^{(1)}}d\boldsymbol{W}_t^{(1)}, \end{aligned} \tag{1}$$

where $\tau^{(P)} > \tau^{(1)}$ and $\boldsymbol{W}_t^{(P)}$ is independent of $\boldsymbol{W}_t^{(1)}$. To promote more explorations for the low-temperature particle, the particles at the position $(\beta^{(1)}, \beta^{(P)}) \in \mathbb{R}^{2d}$ swap with a probability

$$aS(\beta^{(1)}, \beta^{(P)}) = a \cdot \left(1 \wedge e^{\left(\frac{1}{\tau^{(1)}} - \frac{1}{\tau^{(P)}}\right)\left(U(\beta^{(1)}) - U(\beta^{(P)})\right)}\right), \tag{2}$$

where $a \in (0, \infty)$ is the swap intensity. In specific, the conditional swap rate at time $t$ follows that

$$\begin{aligned} \mathbb{P}(\boldsymbol{\beta}_{t+dt} = (\beta^{(P)}, \beta^{(1)})|\boldsymbol{\beta}_t = (\beta^{(1)}, \beta^{(P)})) &= aS(\beta^{(1)}, \beta^{(P)})dt, \\ \mathbb{P}(\boldsymbol{\beta}_{t+dt} = (\beta^{(1)}, \beta^{(P)})|\boldsymbol{\beta}_t = (\beta^{(1)}, \beta^{(P)})) &= 1 - aS(\beta^{(1)}, \beta^{(P)})dt. \end{aligned}$$

In the longtime limit, the Markov jump process converges to the joint distribution $\pi(\boldsymbol{\beta}^{(1)}, \boldsymbol{\beta}^{(P)}) \propto e^{-\frac{U(\boldsymbol{\beta}^{(1)})}{\tau^{(1)}} - \frac{U(\boldsymbol{\beta}^{(P)})}{\tau^{(P)}}}$. For convenience, we refer to the marginal distribution $\pi^{(1)}(\boldsymbol{\beta}) \propto e^{-\frac{U(\boldsymbol{\beta})}{\tau^{(1)}}}$ and $\pi^{(P)}(\boldsymbol{\beta}) \propto e^{-\frac{U(\boldsymbol{\beta})}{\tau^{(P)}}}$ as the *target distribution* and *reference distribution*, respectively.

## 2 PRELIMINARIES

Achieving sufficient explorations requires a large $\tau^{(P)}$, which leads to limited accelerations due to a *small overlap* between $\pi^{(1)}$ and $\pi^{(P)}$. To tackle this issue, one can bring in multiple particles with temperatures $(\tau^{(2)}, \cdots, \tau^{(P-1)})$, where $\tau^{(1)} < \tau^{(2)} < \cdots < \tau^{(P)}$, to hollow out "tunnels". To maintain feasibility, numerous schemes are presented to select candidate pairs to attempt the swaps.

**APE** The all-pairs exchange (APE) attempts to swap arbitrary pair of chains (Brenner et al., 2007; Lingenheil et al., 2009), however, such a method requires a swap time (see definition in section A.5) of $O(P^3)$ and may not be user-friendly in practice.

**ADJ** In addition to swap arbitrary pairs, one can also swap *adjacent* (ADJ) pairs iteratively from $(1, 2)$, $(2, 3)$, to $(P - 1, P)$ under the Metropolis rule. Despite the convenience, the *sequential nature* requires to wait for exchange information from previous exchanges, which only works well with a small number of chains and has greatly limited its extension to a multi-core or distributed context.

**SEO** The stochastic even-odd (SEO) scheme first divides the adjacent pairs $\{(p - 1, p)|p = 2, \cdots, P\}$ into $E$ and $O$, where $E$ and $O$ denote even and odd pairs of forms $(2p - 1, 2p)$ and $(2p, 2p + 1)$, respectively. Then, SEO randomly picks $E$ or $O$ pairs with an equal chance in each iteration to attempt the swaps. Notably, it can be conducted *simultaneously* without waiting from other chains. The scheme yields a reversible process (see Figure 1(a)), however, the gains in overcoming the sequential obstacle don't offset the $O(P^2)$ *round trip time* and SEO is still not effective enough.

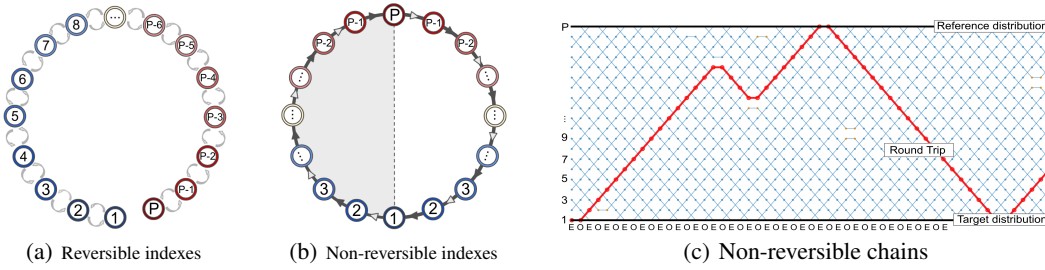

|       |       |       |
|-------|-------|-------|
| (a) Reversible indexes | (b) Non-reversible indexes | (c) Non-reversible chains |

Figure 1: Reversibility v.s. non-reversibility. In (a), a reversible index takes $O(P^2)$ time to communicate; in (b), an ideal non-reversible index moves along a periodic orbit, where the dark and light arrows denote even and odd iterations, respectively; (c) shows how a non-reversible chain conducts a round trip with the DEO scheme.

**DEO** The deterministic even-odd (DEO) scheme instead attempts to swap even ($E$) pairs at even ($E$) iterations and odd ($O$) pairs at odd ($O$) iterations alternatingly[†] (Okabe et al., 2001). The asymmetric manner was later interpreted as a non-reversible PT (Syed et al., 2021) and an ideal index process follows a periodic orbit, as shown in Figure 1(b). With a large swap rate, Figure 1(c) shows how the scheme yields an almost straight path and a linear round trip time can be expected.

**Equi-acceptance** The power of PT hinges on maximizing the number of round trips, which is equivalent to minimizing $\sum_{p=1}^{P-1} \frac{1}{1-r_p}$ (Nadler & Hansmann, 2007b), where $r_p$ denotes the rejection rate for the chain pair $(p, p + 1)$. Moreover, $\sum_{p=1}^{P-1} r_p$ converges to a fixed barrier $\Lambda$ as $P \to \infty$ (Predescu et al., 2004; Syed et al., 2021). Applying Lagrange multiplies to the constrained optimization problem leads to $r_1 = r_2 = \cdots = r_{P-1} := r$, where $r$ is the *equi-rejection rate*. In general, a quadratic round trip time is required for ADJ and SEO due to the reversible indexes. By contrast, DEO only yields a *linear round trip* time in terms of $P$ as $P \to \infty$ Syed et al. (2021).

## 3 OPTIMAL NON-REVERSIBLE SCHEME FOR PARALLEL TEMPERING

The linear round trip time is appealing for maximizing the algorithmic potential, however, such an advance only occurs given sufficiently many chains. In *non-asymptotic settings* with limited chains, a pearl of wisdom is to avoid frequent swaps (Dupuis et al., 2012) and to keep the average acceptance rate from 20% to 40% (Kone & Kofke, 2005; Lingenheil et al., 2009; Atchadé et al., 2011).

---

[†] $E$ shown in iterations means even iterations; otherwise, it denotes even pairs for chain indexes. The same logic applies to $O$.

Most importantly, the acceptance rates are severely reduced in big data due to the bias-corrected swaps associated with stochastic energies (Deng et al., 2020), see details in section A.1. As such, maintaining low rejection rates becomes quite challenging and the *issue of quadratic costs* still exists.

### 3.1 GENERALIZED DEO SCHEME

Continuing the equi-acceptance settings, we see in Figure.2(a) that the probability for the blue particle to move upward 2 steps to maintain the same momentum after a pair of even and odd iterations is $(1 - r)^2$. As such, with a large equi-rejection rate $r$, the blue particle often makes little progress (Figure.2(b-d)). To handle this issue, the key is to propose small enough rejection rates to track the periodic orbit in Figure.1(b). Instead of pursuing excessive amount of chains, *we resort to a different solution by introducing the generalized even and odd iterations $E_W$ and $O_W$, where $W \in \mathbb{N}^+$,* $E_W = \{\lfloor \frac{k}{W} \rfloor \mod 2 = 0 | k = 1, 2, \cdots, \infty\}$ and $O_W = \{\lfloor \frac{k}{W} \rfloor \mod 2 = 1 | k = 1, 2, \cdots, \infty\}$. Now, we present the generalized DEO scheme with a window size $W$ as follows and refer to it as DEO$_W$:[§]

> ○ Attempt to swap $E$ (or $O$) pairs at $E_W$ (or $O_W$) iterations.
> ○ Allow *at most one* swap during each cycle of $E_W$ (or $O_W$) iterations.

(3)

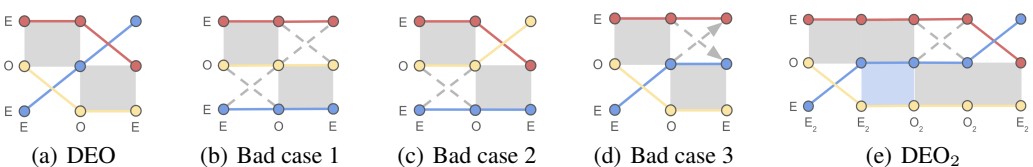

|     (a) DEO     |  (b) Bad case 1  |  (c) Bad case 2  |  (d) Bad case 3  |   (e) DEO$_2$   |

Figure 2: Illustration of DEO and DEO$_2$. In (a), we show an ideal case of DEO; (b-d) show bad cases of DEO based on a large equi-rejection rate $r$; (e) show how the generalized DEO scheme of window size 2 tackles the issue with a large $r$. The x-axis and y-axis denote (generalized) $E$ (or $O$) iterations and $E$ (or $O$) pairs, respectively. The dashed line denotes the failed swap attempts; the gray shaded areas are frozen to refuse swapping odd pairs at even iterations (or vice versa); the blue shaded area ensures *at most one swap in a window*.

As illustrated in Figure.2(e), the blue particle has a larger chance of $(1 - r^2)^2$ to move upward 2 steps given $W = 2$ instead of $(1 - r)^2$ when $W = 1$, although the window number is also halved. Such a trade-off inspires us to analyze the expected round trip time based on the window of size $W$. Although allowing at most one swap introduces the stopping time and may affect the distribution, the bias is rather mild due to the noisy energy estimators in big data. Check section C.2 for the details.

### 3.2 ANALYSIS OF ROUND TRIP TIME

To bring sufficient interactions between the reference distribution $\pi^{(P)}$ and the target distribution $\pi^{(1)}$, we expect to minimize the expected round trip time $T$ (defined in section A.5) to ensure both efficient exploitation and explorations. Combining the Markov property and the idea of the master equation (Nadler & Hansmann, 2007a), we estimate the expected round trip time $\mathbb{E}[T]$ as follows

**Lemma 1.** *Under the stationary and weak dependence assumptions B1 and B2 in section B, for $P$ ($P \geq 2$) chains with window size $W$ ($W \geq 1$) and rejection rates $\{r_p\}_{p=1}^{P-1}$, we have*

$$\mathbb{E}[T] = 2WP + 2WP \sum_{p=1}^{P-1} \frac{r_p^W}{1 - r_p^W}. \tag{4}$$

The proof in section B.1 shows that $\mathbb{E}[T]$ increases as we adopt larger number of chains $P$ and rejection rates $\{r_p\}_{p=1}^{P-1}$. In such a case, the round trip rate $\frac{P}{\mathbb{E}[T]}$ is also maximized by the key renewal theorem. In particular, applying $W = 1$ recovers the vanilla DEO scheme.

### 3.3 ANALYSIS OF OPTIMAL WINDOW SIZE AND ROUND TRIP TIME

By Lemma 1, we observe a potential to remove the second quadratic term given an appropriate $W$. Such a fact motivates us to study the optimal window size $W$ to achieve the best efficiency. Under

---

[§]The generalized DEO with the optimal window size is denoted by DEO$_\star$ and will be studied in section 3.3.

the equi-acceptance settings, by treating the window size $W$ as a continuous variable and taking the derivative of $\mathbb{E}[T]$ with respect to $W$, we have

$$\frac{\partial}{\partial W}\mathbb{E}[T] = \frac{2P}{(1-r^W)^2}\left\{(1-r^W)^2 + (P-1)r^W(1-r^W + W\log r)\right\}, \qquad (5)$$

where $r$ is the equi-rejection rate for adjacent chains. Define $x := r^W \in (0,1)$, where $W = \log_r(x) = \frac{\log x}{\log r}$. The following analysis hinges on the study of the solution $g(x) = (1-x)^2 + (P-1)x(1-x+\log(x)) = 0$. By analyzing the growth of derivatives and boundary values, we can easily identify the *uniqueness* of the solution. Then, we proceed to verify that $\frac{1}{P\log P}$ yields an asymptotic approximation such that $g(\frac{1}{P\log P}) = -\frac{\log(\log P)}{\log P} + O\left(\frac{1}{\log P}\right) \to 0$ as $P \to \infty$. In the end, we have

**Theorem 1.** *Under Assumptions B1 and B2 based on equi-acceptance settings, if $P = 2, 3$, the maximal round trip time is achieved when $W = 1$. If $P \geq 4$, with the optimal window size $W_\star \approx \left\lceil \frac{\log P + \log\log P}{-\log r} \right\rceil$, where $\lceil\cdot\rceil$ is the ceiling function. The round trip time follows $O(\frac{P\log P}{-\log r})$.*

The above result yields a remarkable round trip time of $O(P\log P)$ by setting the optimal window size $W_\star$. By contrast, the vanilla DEO scheme only leads to a longer time of $O(P^2)$ [§]. Denoting by DEO$_\star$ the generalized DEO scheme with the optimal window size $W_\star$, we summarize the popular swap schemes in Table.1, where the DEO$_\star$ scheme performs the best among all the three criteria.

TABLE 1: ROUND TRIP TIME AND SWAP TIME FOR DIFFERENT SCHEMES. THE APE SCHEME REQUIRES AN EXPENSIVE SWAP TIME OF $O(P^3)$ AND IS NOT COMPARED.

| | ROUND TRIP TIME (NON-ASYMPTOTIC) | ROUND TRIP TIME (ASYMPTOTIC) | SWAP TIME |
|---|---|---|---|
| ADJ | $O(P^2)$ (NADLER & HANSMANN, 2007A) | $O(P^2)$ (NADLER & HANSMANN, 2007A) | $O(P)$ |
| SEO | $O(P^2)$ (SYED ET AL., 2021) | $O(P^2)$ (SYED ET AL., 2021) | $O(1)$ |
| DEO | $O(P^2)$ (SYED ET AL., 2021) | $O(P)$ (SYED ET AL., 2021) | $O(1)$ |
| DEO$_\star$ | $O(P\log P)$ | $O(P)$ | $O(1)$ |

### 3.4 DISCUSSIONS ON THE OPTIMAL NUMBER OF CHAINS

Note that in practice given $P$ parallel chains, a large $P$ leads to a smaller equi-rejection rate $r$. As such, we can further obtain a crude estimate of the optimal $P$ to minimize the round trip time.

**Corollary 1.** *Under Assumptions B1-B4 and C1 under equi-acceptance settings with the optimal window size, the optimal number of chains follows that $P_\star > \min_p \frac{\sigma_p}{3\tau^{(p)}}\log(\frac{\tau^{(P)}}{\tau^{(1)}})$, where $\sigma_p$ is defined in Eq.(15).*

The assumptions and proof are postponed in section B.3. In mini-batch settings, insufficient chains may lead to few effective swaps for accelerations; by contrast, introducing too many chains may be too costly in terms of the round trip time. This is different from the conclusion in full-batch settings, where Syed et al. (2021) suggested running the vanilla DEO scheme with as many chains as possible to yield a small enough equi-rejection rate $r$ to maintain the non-reversibility.

**Cutoff phenomenon** On the one hand, when we only afford at most $P$ chains, where $P < P_\star$, a large equi-rejection rate $r$ is inevitable and DEO$_\star$ is preferred over DEO; on the other hand, the rejection rate $r$ goes to 0 when $P \gg P_\star$ and DEO$_\star$ recovers the DEO scheme.

In section B.4, we show that $P_\star$ is in the *order of thousands* for the CIFAR100 example, which is hard to achieve due to the limited computational budget and further motivates us to adopt finite chains with a target swap rate $\mathbb{S}$ to balance between acceleration and accuracy.

## 4 USER-FRIENDLY APPROXIMATE EXPLORATIONS IN BIG DATA

Despite the asymptotic correctness, SGLD only works well given *small enough learning rates* and fails in explorative purposes (Ahn et al., 2012). A large learning rate, however, leads to excessive stochastic gradient noise and ends up with a crude approximation. As such, similar to Izmailov et al. (2018); Zhang et al. (2020), we only adopt SGLD for exploitations.

---

[§]By Taylor expansion, given a large rejection rate $r$, $-\log(r) = 1 - r$, which means $\frac{1}{-\log(r)} = O(\frac{r}{1-r})$.

Efficient explorations not only require a high temperature but also prefer a large learning rate. Such a demand inspires us to consider SGD with a constant learning rate $\eta$ as the exploration component

$$\boldsymbol{\beta}_{k+1} = \boldsymbol{\beta}_k - \eta(\nabla U(\boldsymbol{\beta}_k) + \varepsilon(\boldsymbol{\beta}_k)) = \boldsymbol{\beta}_k - \eta\nabla U(\boldsymbol{\beta}_k) + \sqrt{2\eta\left(\frac{\eta}{2}\right)}\varepsilon(\boldsymbol{\beta}_k), \quad (6)$$

where $\varepsilon(\boldsymbol{\beta}_k) \in \mathbb{R}^d$ is the stochastic gradient noise. Under mild normality assumptions on $\varepsilon$ (Mandt et al., 2017; Chen et al., 2020), $\boldsymbol{\beta}_k$ converges approximately to an invariant distribution, where the underlying *temperature linearly depends on the learning rate* $\eta$. Motivated by this fact, we propose an approximate transition kernel $\mathcal{T}_\eta$ with $P$ parallel *SGD runs* based on different learning rates

$$
\textbf{Exploration:} \begin{cases}
\boldsymbol{\beta}_{k+1}^{(P)} = \boldsymbol{\beta}_k^{(P)} - \eta^{(P)}\nabla\widetilde{U}(\boldsymbol{\beta}_k^{(P)}), \\
\cdots \\
\boldsymbol{\beta}_{k+1}^{(2)} = \boldsymbol{\beta}_k^{(2)} - \eta^{(2)}\nabla\widetilde{U}(\boldsymbol{\beta}_k^{(2)}),
\end{cases} \quad (7)
$$

$$\textbf{Exploitation:} \quad \boldsymbol{\beta}_{k+1}^{(1)} = \boldsymbol{\beta}_k^{(1)} - \eta^{(1)}\nabla\widetilde{U}(\boldsymbol{\beta}_k^{(1)}) + \overbrace{\Xi_k}^{\text{optional}},$$

where $\eta^{(1)} < \eta^{(2)} < \cdots < \eta^{(P)}$, $\Xi_k \sim \mathcal{N}(0, 2\eta^{(1)}\tau^{(1)})$, and $\tau^{(1)}$ is the target temperature.

Since there exists an optimal learning rate for SGD to estimate the desired distribution through Laplace approximation (Mandt et al., 2017), the exploitation kernel can be also replaced with SGD based on constant learning rates if the accuracy demand is not high. Regarding the validity of adopting different learning rates for parallel tempering, we leave discussions to section A.2.

## 4.1 APPROXIMATION ANALYSIS

Moreover, the stochastic gradient noise exploits the Fisher information (Ahn et al., 2012; Zhu et al., 2019; Chaudhari et al., 2017) and yields convergence potential to wide optima with good generalizations (Berthier et al., 2020; Zou et al., 2021). Despite the implementation convenience, the inclusion of SGDs has made the temperature variable inaccessible, rendering a difficulty in implementing the Metropolis rule Eq.(2). To tackle this issue, we utilize the randomness in stochastic energies and propose a *deterministic swap condition* for the approximate kernel $\mathcal{T}_\eta$ in Eq.(7) such that

**Deterministic swap condition:** $(\boldsymbol{\beta}^{(p)}, \boldsymbol{\beta}^{(p+1)}) \to (\boldsymbol{\beta}^{(p+1)}, \boldsymbol{\beta}^{(p)})$ if $\widetilde{U}(\boldsymbol{\beta}^{(p+1)}) + \mathbb{C} < \widetilde{U}(\boldsymbol{\beta}^{(p)})$, (8)

where $p \in \{1, 2, \cdots, P-1\}$, $\mathbb{C} > 0$ is a correction buffer to approximate the Metropolis rule Eq.(2).

**Lemma 2.** *Assume the energy normality assumption (C1), then for any fixed $\partial U_p := U(\boldsymbol{\beta}^{(p)}) - U(\boldsymbol{\beta}^{(p+1)})$, there exists an optimal $\mathbb{C}_\star \in (0, (\frac{1}{\tau^{(p)}} - \frac{1}{\tau^{(p+1)}})\sigma_p^2]$ that perfectly approximates the random event $\widetilde{S}(\boldsymbol{\beta}^{(p)}, \boldsymbol{\beta}^{(p+1)}) > u$, where $\sigma_p$ defined in Eq.(15) and $u \sim Unif\,[0, 1]$.*

The proof is postponed in section C.1, which paves the way for the guarantee that a *deterministic swap condition* may replace the Metropolis rule Eq.(2) for approximation tasks. In addition, the normality assumption can be naturally extended to the asymptotic normality assumption (Quiroz et al., 2019; Deng et al., 2021) given large enough batch sizes.

Admittedly, the approximation error still exists for different $\partial U_p$. By the mean-value theorem, there exists a tunable $\mathbb{C}$ to optimize the overall approximation. Further invoking the central limit theorem such that $\varepsilon(\cdot)$ in Eq.(6) approximates a Gaussian distribution with a fixed covariance, we can expect a bounded approximation error for the SGD-based exploration kernels (Mandt et al., 2017).

**Theorem 2.** *Consider the exact transition kernel $\mathcal{T}$ and the proposed approximate kernel $\mathcal{T}_\eta$, which yield stationary distributions $\pi$ and $\pi_\eta$, respectively. Under smoothness (C2) and dissipativity assumptions (C3) (Mattingly et al., 2002; Raginsky et al., 2017; Xu et al., 2018), $\mathcal{T}$ satisfies the geometric ergodicity such that there is a contraction constant $\rho \in [0, 1)$ for any distribution $\mu$:*

$$\|\mu\mathcal{T} - \pi\|_{TV} \leq \rho\|\mu - \pi\|_{TV},$$

*where $\|\cdot\|_{TV}$ is the total variation (TV) distance. Moreover, assume that $\varepsilon(\cdot) \sim \mathcal{N}(0, \mathcal{M})$ for some positive definite matrix $\mathcal{M}$ (C4) (Mandt et al., 2017), then there is a uniform upper bound of the one step error between $\mathcal{T}$ and $\mathcal{T}_\eta$ such that*

$$\|\mu\mathcal{T} - \mu\mathcal{T}_\eta\|_{TV} \leq \Delta_{\max}, \forall\mu,$$

where $\Delta_{\max} \geq 0$ is a constant. Eventually, the TV distance between $\pi$ and $\pi_\eta$ is bounded by

$$\|\pi - \pi_\eta\|_{TV} \leq \frac{\Delta_{\max}}{1 - \rho}.$$

The proof is postponed to section C.2. The SGD-based exploration kernels *no longer require to fine-tune the temperatures* directly and naturally inherits the empirical successes of SGD in large-scale deep learning tasks. The inaccessible Metropolis rule Eq.(2) is approximated via the *deterministic swap condition* Eq.(8) and leads to a well-controlled approximations by *solely tuning* $\boldsymbol{\eta} = (\eta^{(1)}, \cdots, \eta^{(P)})$ *and* $\mathbb{C}$.

In addition, our proposed algorithm for uncertainty approximation is highly related to non-convex optimization. For the detailed discussions, we refer interested readers to section A.4.

## 4.2 EQUI-ACCEPTANCE PARALLEL TEMPERING ON OPTIMIZED PATHS

Stochastic approximation (SA) is a standard method to achieve equi-acceptance (Atchadé et al., 2011; Miasojedow et al., 2013), however, implementing this idea with fixed $\eta^{(1)}$ and $\eta^{(P)}$ is rather non-trivial. Motivated by the linear relation between learning rate and temperature, we propose to adaptively *optimize the learning rates* to achieve equi-acceptance in a user-friendly manner. Further by the geometric temperature spacing commonly adopted by practitioners (Kofke, 2002; Earl & Deem, 2005; Syed et al., 2021), we adopt the following scheme on a *logarithmic scale* such that

$$\partial \log(\upsilon_t^{(p)}) = h^{(p)}(\upsilon_t^{(p)}), \tag{9}$$

where $p \in \{1, 2, \cdots, P-1\}$, $\upsilon_t^{(p)} = \eta_t^{(p+1)} - \eta_t^{(p)}$, $h^{(p)}(\upsilon_t^{(p)}) := \int H^{(p)}(\upsilon_k^{(p)}, \boldsymbol{\beta}) \pi^{(p,p+1)}(d\boldsymbol{\beta})$ is the mean-field function, $\pi^{(p,p+1)}$ is the joint invariant distribution for the $p$-th and $p+1$-th processes. In particular, $H^{(p)}(\upsilon_k^{(p)}, \boldsymbol{\beta}) = 1_{\widetilde{U}(\boldsymbol{\beta}^{(p+1)}) + \mathbb{C} < \widetilde{U}(\boldsymbol{\beta}^{(p)})} - \mathbb{S}$ is the random-field function to approximate $h^{(p)}(\upsilon_k^{(p)})$ [†] with limited perturbations, $\upsilon_k^{(p)}$ implicitly affects the distribution of the indicator function, and $\mathbb{S}$ is the target swap rate. Now consider stochastic approximation of Eq.(9), we have

$$\log(\upsilon_{k+1}^{(p)}) = \log(\upsilon_k^{(p)}) + \gamma_k H^{(p)}(\upsilon_k^{(p)}, \boldsymbol{\beta}_k), \tag{10}$$

where $\gamma_k$ is the step size. Reformulating Eq.(10), we have

$$\upsilon_{k+1}^{(p)} = \max(0, \upsilon_k^{(p)}) e^{\gamma_k H^{(p)}(\upsilon_k^{(p)})},$$

where the $\max$ operator is conducted explicitly to ensure the sequence of learning rates is non-decreasing. This means that given fixed boundary learning rates (temperatures) $\eta_k^{(p-1)}$ and $\eta_k^{(p+1)}$, applying $\eta^{(p)} = \eta^{(p-1)} + \upsilon^{(p)}$ and $\eta^{(p)} = \eta^{(p+1)} - \upsilon^{(p+1)}$ for $p \in \{2, 3, \cdots, P-1\}$ lead to

$$\underbrace{\eta_k^{(p-1)} + \max(0, \upsilon_k^{(p)}) e^{\gamma_k H(\upsilon_k^{(p)})}}_{\text{forward sequence}} = \eta_{k+1}^{(p)} = \underbrace{\eta_k^{(p+1)} - \max(0, \upsilon_k^{(p+1)}) e^{\gamma_k H(\upsilon_k^{(p+1)})}}_{\text{backward sequence}}. \tag{11}$$

**Adaptive learning rates (temperatures)**  Now given a fixed $\eta^{(1)}$, the sequence $\eta^{(2)}, \eta^{(3)}, \cdots, \eta^{(P)}$ can be approximated iteratively via the forward sequence of (11); conversely, given a fixed $\eta^{(P)}$, the backward sequence $\eta^{(P-1)}, \eta^{(P-2)}, \cdots, \eta^{(1)}$ can be decided reversely as well. Combining the forward and backward sequences, $\eta_{k+1}^{(p)}$ can be approximated via

$$\eta_{k+1}^{(p)} := \frac{\eta_k^{(p-1)} + \eta_k^{(p+1)}}{2} + \frac{\max(0, \upsilon_k^{(p)}) e^{\gamma_k H(\upsilon_k^{(p)})} - \max(0, \upsilon_k^{(p+1)}) e^{\gamma_k H(\upsilon_k^{(p+1)})}}{2}, \tag{12}$$

which resembles the *binary search* in the SA framework. In particular, the first term is the middle point given boundary learning rates and the second term continues to penalize learning rates that violates the equi-acceptance between pairs $(p-1, p)$ and $(p, p+1)$ until an equilibrium is achieved.

This is the first attempt to achieve equi-acceptance given two fixed boundary values to our best knowledge. By contrast, Syed et al. (2021) proposed to estimate the barrier $\Lambda$ to determine the temperatures and it easily fails in big data given a finite number of chains and bias-corrected swaps.

---

[†]For convenience, $\upsilon_t^{(p)}$ denotes the continuous-time diffusion at time $t$ and $\upsilon_k^{(p)}$ represents the discrete approximations at iteration $k$.

---

**Algorithm 1** Non-reversible parallel tempering with SGD-based exploration kernels (DEO$_\star$-SGD).

---

**Input** Number of chains $P \geq 3$, boundary learning rates $\eta^{(1)}$ and $\eta^{(P)}$, target swap rate $\mathbb{S}$.

**Input** Optimal window size $W := \left\lceil \frac{\log P + \log \log P}{-\log(1-\mathbb{S})} \right\rceil$, total iterations $K$, and step sizes $\{\gamma_k\}_{k=0}^K$.

**for** $k = 1$ **to** $K$ **do**
  $\boldsymbol{\beta}_{k+1} \sim \mathcal{T}_\eta(\boldsymbol{\beta}_k)$ following Eq.(7)            $\triangleright$ Exploration / exploitation phase (parallelizable)
  $\mathcal{P} = \{\forall p \in \{1, 2, \cdots, P\} : p \bmod 2 = \lfloor \frac{k}{W} \rfloor \bmod 2\}$.            $\triangleright$ Generalized even/odd iterations
  **for** $p = 1, 2$ **to** $P - 1$ **do**
    $\mathcal{A}^{(p)} := 1_{\widetilde{U}(\boldsymbol{\beta}_{k+1}^{(p+1)}) + \mathbb{C}_k < \widetilde{U}(\boldsymbol{\beta}_{k+1}^{(p)})}$
    $\mathcal{G}^{(p)} := 1_{k \bmod W = 0}$.            $\triangleright$ Open the gate to allow swaps
    **if** $p \in \mathcal{P}$ **and** $\mathcal{G}^{(p)}$ **and** $\mathcal{A}^{(p)}$ **then**
      *Swap:* $\boldsymbol{\beta}_{k+1}^{(p)}$ and $\boldsymbol{\beta}_{k+1}^{(p+1)}$.            $\triangleright$ Communication phase (parallelizable)

      *Freeze:* $\mathcal{G}^{(p)} = 0$.            $\triangleright$ Close the gate to refuse swaps
    **end if**
    **if** $p > 1$ **then**
      *Update learning rate (temperature) following Eq.(12)*
    **end if**
  **end for**
  *Adaptive correction buffer:* $\mathbb{C}_{k+1} = \mathbb{C}_k + \gamma_k \left( \frac{1}{P-1} \sum_{p=1}^{P-1} \mathcal{A}^{(p)} - \mathbb{S} \right)$.
**end for**
**Output** Models collected from the target temperature $\{\boldsymbol{\beta}_k^{(1)}\}_{k=1}^K$.

---

**Adaptive correction buffers**  In addition, equi-acceptance does not guarantee a convergence to the desired acceptance rate $\mathbb{S}$. To avoid this issue, we propose to adaptively optimize $\mathbb{C}$ as follows

$$\mathbb{C}_{k+1} = \mathbb{C}_k + \gamma_k \left( \frac{1}{P-1} \sum_{p=1}^{P-1} 1_{\widetilde{U}(\boldsymbol{\beta}_{k+1}^{(p+1)}) + \mathbb{C}_k - \widetilde{U}(\boldsymbol{\beta}_{k+1}^{(p)}) < 0} - \mathbb{S} \right). \tag{13}$$

As $k \to \infty$, the threshold and the adaptive learning rates converge to the desired fixed points. Note that setting a uniform $\mathbb{C}$ greatly simplifies the algorithm; in more delicate cases, problem-specific rules are also recommended. Now we refer to the approximate non-reversible parallel tempering algorithm with the DEO$_\star$ scheme and SGD-based exploration kernels as DEO$_\star$-SGD and formally formulate our algorithm in Algorithm 1. Extensions of SGD with a preconditioner (Li et al., 2016) or momentum (Chen et al., 2014) to further improve the approximation and efficiency are both straightforward (Mandt et al., 2017) and are denoted as DEO$_\star$-pSGD and DEO$_\star$-mSGD, respectively.

## 5 EXPERIMENTS

### 5.1 SIMULATIONS OF MULTI-MODAL DISTRIBUTIONS

We first simulate the proposed algorithm on a distribution $\pi(\boldsymbol{\beta}) \propto \exp(-U(\boldsymbol{\beta}))$, where $\boldsymbol{\beta} = (\beta_1, \beta_2)$, $U(\boldsymbol{\beta}) = 0.2(\beta_1^2 + \beta_2^2) - 2(\cos(2\pi\beta_1) + \cos(2\pi\beta_2))$. The heat map is shown in Figure 3(a) with 25 modes of different volumes. To mimic big data scenarios, we can only access stochastic gradient $\nabla\widetilde{U}(\boldsymbol{\beta}) = \nabla U(\boldsymbol{\beta}) + 2\mathcal{N}(0, \boldsymbol{I}_{2\times2})$ and stochastic energy $\widetilde{U}(\boldsymbol{\beta}) = U(\boldsymbol{\beta}) + 2\mathcal{N}(0, I)$.

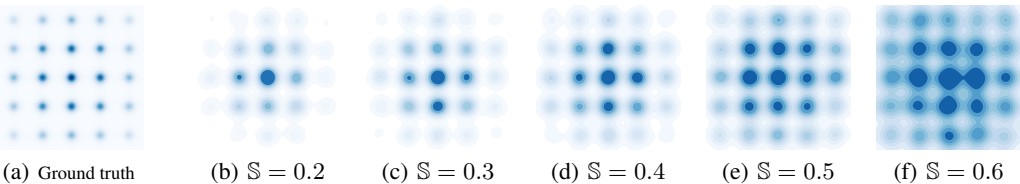

(a) Ground truth    (b) $\mathbb{S} = 0.2$    (c) $\mathbb{S} = 0.3$    (d) $\mathbb{S} = 0.4$    (e) $\mathbb{S} = 0.5$    (f) $\mathbb{S} = 0.6$

Figure 3: Study of different target swap rate $\mathbb{S}$ via DEO$_\star$-SGD, where SGLD is the exploitation kernel.

We first run DEO$_\star$-SGD$\times$P16 based on 16 chains and 20,000 iterations. We fix the lowest learning rate 0.003 and the highest learning 0.6 and propose to tune the target swap rate $\mathbb{S}$ for the acceleration-accuracy trade-off. Fig.3 shows that fixing $\mathbb{S} = 0.2$ or 0.3 is too conservative and underestimates the

uncertainty on the corners; $\mathbb{S} = 0.6$ results in too many radical swaps and eventually leads to crude estimations; by contrast, $\mathbb{S} = 0.4$ yields the best uncertainty approximation among the five choices.

Next, we select $\mathbb{S} = 0.4$ and study the round trips. We observe in Fig.4(a) that the vanilla DEO only yields 18 round trips every 1,000 iterations; by contrast, slightly increasing $W$ tends to improve the efficiency significantly and the optimal 45 round trips are achieved at $W = 8$, which *matches our theory*. In Fig.4(b-c), the geometrically initialized learning rates lead to unbalanced acceptance rates in the early phase and some adjacent chains have few swaps and others swap too much, but as the optimization proceeds, the learning rates gradually converge. We also observe in Fig.4(d) that the correction is adaptively estimated to ensure the average acceptance rates converge to $\mathbb{S} = 0.4$.

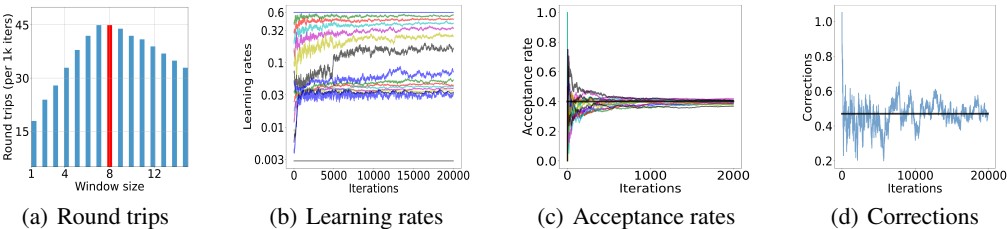

| (a) Round trips | (b) Learning rates | (c) Acceptance rates | (d) Corrections |

Figure 4: Study of window sizes, learning rates, acceptance rates, and the corrections.

We compare the proposed algorithm with parallel SGLD based on 20,000 iterations and 16 chains (SGLD×P16); we fix the learning rate 0.003 and a temperature 1. We also run cycSGLD×T16, which is short for a single long chain based on 16 times of budget and cosine learning rates (Zhang et al., 2020) of 100 cycles. We see in Figure 5(b) that SGLD×P16 has good explorations but fails to quantify the uncertainty. Figure 5(c) shows that cycSGLD×T16 explores most of the modes but overestimates some areas occasionally. Figure 5(d) demonstrates the DEO-SGD with 16 chains (DEO-SGD×P16) estimates the uncertainty of the centering 9 modes well but fails to deal with the rest of the modes. As to DEO$_\star$-SGD×P16, the approximation is rather accurate, as shown in Fig.5(e).

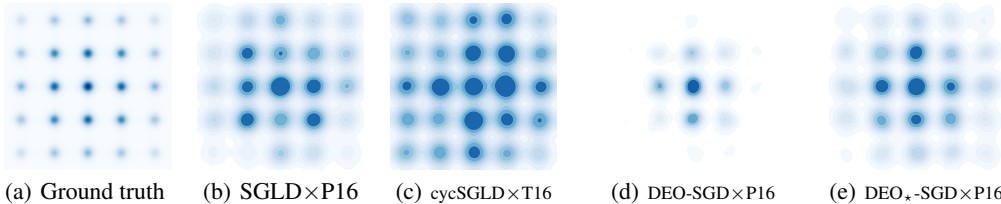

| (a) Ground truth | (b) SGLD×P16 | (c) cycSGLD×T16 | (d) DEO-SGD×P16 | (e) DEO$_\star$-SGD×P16 |

Figure 5: Simulations of the multi-modal distribution through different sampling algorithms.

We also present the index process for both schemes in Fig.6. We see that the vanilla DEO scheme results in volatile paths and a particle takes quite a long time to complete a round trip; by contrast, DEO$_\star$ only conducts at most one cheap swap in a window and yields much more deterministic paths.

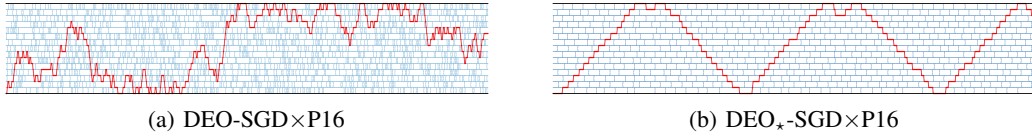

| (a) DEO-SGD×P16 | (b) DEO$_\star$-SGD×P16 |

Figure 6: Dynamics of the index process. The red path denotes the round trip path for a particle.

## 5.2 UNCERTAINTY APPROXIMATION AND OPTIMIZATION FOR IMAGE DATA

Next, we conduct experiments on computer vision tasks. We choose ResNet20, ResNet32, and ResNet56 (He et al., 2016) and train the models on CIFAR100. We not only report test negative log likelihood (NLL) but also present the test accuracy (ACC). For each ResNet model, we first pre-train 10 fixed models via 300 epochs and then run our algorithm based on momentum SGD (mSGD) for 500 epochs with 10 parallel chains and denote it by DEO$_\star$-mSGD×P10. We fix the lowest and highest learning rates as 0.005 and 0.02, respectively. For a fair comparison, we also include the baseline DEO-mSGD×P10 with the same setup except that the window size is 1; the standard ensemble mSGD×P10 is also included with a learning rate of 0.005. In addition, we include two baselines based on a single long chain, i.e. we run stochastic gradient Hamiltonian Monte Carlo

(Chen et al., 2014) 5000 epochs with cyclical learning rates and 50 cycles (Zhang et al., 2020) and refer to it as cycSGHMC×T10; we run SWAG×T10 (Maddox et al., 2019) under a similar setup.

In particular for DEO$_\star$-mSGD×P10, we tune the target swap rate $\mathbb{S}$ and find an optimum at $\mathbb{S} = 0.005$. We compare our proposed algorithm with the four baselines and observe in Table.2 that mSGD×P10 can easily obtain competitive results simply through model ensemble (Lakshminarayanan et al., 2017), which outperforms cycSGHMC×T10 and cycSWAG×T10 on ResNet20 and ResNet32 models and perform the worst among the five methods on ResNet56; DEO-mSGD×P10 itself is already a pretty powerful algorithm, however, DEO$_\star$-mSGD×P10 consistently outperforms the vanilla alternative.

TABLE 2: UNCERTAINTY APPROXIMATION AND OPTIMIZATION ON CIFAR100 VIA 10× BUDGET.

| MODEL | ResNet20 | | ResNet32 | | ResNet56 | |
|---|---|---|---|---|---|---|
| | NLL | ACC (%) | NLL | ACC (%) | NLL | ACC (%) |
| cycSGHMC×T10 | 8198±59 | 76.26±0.18 | 7401±28 | 78.54±0.15 | 6460±21 | 81.78±0.08 |
| cycSWAG×T10 | 8164±38 | 76.13±0.21 | 7389±32 | 78.62±0.13 | 6486±29 | 81.60±0.14 |
| mSGD×P10 | 7902±64 | 76.59±0.11 | 7204±29 | 79.02±0.09 | 6553±15 | 81.49±0.09 |
| DEO-mSGD×P10 | 7964±23 | 76.84±0.12 | 7152±41 | 79.34±0.15 | 6534±26 | 81.72±0.12 |
| DEO$_\star$-mSGD×P10 | **7741±67** | **77.37±0.16** | **7019±35** | **79.54±0.12** | **6439±32** | **82.02±0.15** |

To analyze why the proposed scheme performs well, we study the round trips in Figure.7(a) and find that the theoretical optimal window obtains around 11 round trips every 100 epochs, which is almost 2 times as much as the vanilla DEO scheme. In Figure 7(b), we observe that the smallest learning rate obtains the highest accuracy (blue) for exploitations, while the largest learning rate yields decent explorations (red); we see in Figure 7(c-d) that geometrically initialized learning rate fails in producing equi-acceptance, but as the training proceeds, the learning rates converge to fixed points and the acceptance rates for different pairs converge to the target swap rate.

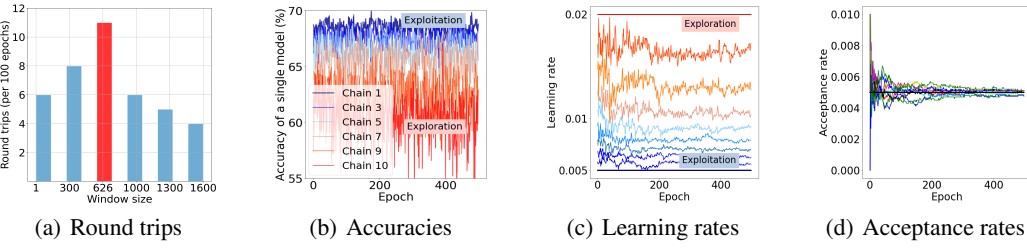

| (a) Round trips | (b) Accuracies | (c) Learning rates | (d) Acceptance rates |

Figure 7: Study of window sizes, accuracies, learning rates, and acceptance rates on ResNet20.

For the visualization of the index process, we observe in Figure.8(a) that the vanilla DEO scheme leads to volatile trajectories wandering back and forth and is rather inefficient; by contrast, the DEO$_\star$ scheme yields well-motivated paths with more deterministic round trips. Interestingly, this path resembles cyclic learning rates (Zhang et al., 2020), which provides a novel viewpoint to *interpret why cyclic learning rates work well empirically*. Nevertheless, the stochastic and parallel manner further improves the margin and eventually leads to the most efficient approximations in this task.

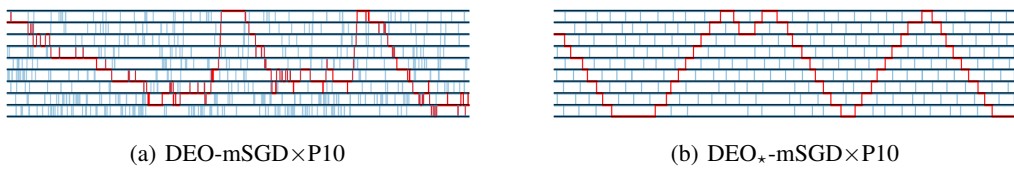

| (a) DEO-mSGD×P10 | (b) DEO$_\star$-mSGD×P10 |

Figure 8: Dynamics of the index process of ResNet20 models in the last 200 epochs.

## 6 CONCLUSION

In this paper, we show how to conduct efficient PT in big data problems. To tackle the inefficiency issue of the popular DEO scheme given limited chains, we present the DEO$_\star$ scheme by applying an optimal window size to encourage *deterministic paths* and obtain in a significant *acceleration of* $O(\frac{P}{\log P})$ *times*. For a user-friendly purpose, we propose a deterministic swap condition to interact with SGD-based exploration kernels and provide a theoretical guarantee to control the bias solely depending on the learning rate $\eta$ and the correction buffer $\mathbb{C}$; we also provide a practical algorithm to adaptively approximate $\eta$ and $\mathbb{C}$ for achieving the optimal efficiency in *constrained settings*.

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

# A BACKGROUND

## A.1 REPLICA EXCHANGE STOCHASTIC GRADIENT LANGEVIN DYNAMICS

To approximate the replica exchange Langevin diffusion Eq.(1) based on multiple particles $(\boldsymbol{\beta}_{k+1}^{(1)}, \boldsymbol{\beta}_{k+1}^{(2)}, \cdots, \boldsymbol{\beta}_{k+1}^{(P)})$ in big data scenarios, replica exchange stochastic gradient Langevin dynamics (reSGLD) proposes the following numerical scheme:

$$
\text{Exploration:} \begin{cases}
\boldsymbol{\beta}_{k+1}^{(P)} = \boldsymbol{\beta}_k^{(P)} - \eta^{(P)} \nabla \widetilde{U}(\boldsymbol{\beta}_k^{(P)}) + \sqrt{2\eta^{(P)}\tau^{(P)}} \boldsymbol{\xi}_k^{(P)}, \\
\cdots \\
\boldsymbol{\beta}_{k+1}^{(2)} = \boldsymbol{\beta}_k^{(2)} - \eta^{(2)} \nabla \widetilde{U}(\boldsymbol{\beta}_k^{(2)}) + \sqrt{2\eta^{(2)}\tau^{(2)}} \boldsymbol{\xi}_k^{(2)}, \\
\end{cases}
$$
$$
\text{Exploitation:} \quad \boldsymbol{\beta}_{k+1}^{(1)} = \boldsymbol{\beta}_k^{(1)} - \eta^{(1)} \nabla \widetilde{U}(\boldsymbol{\beta}_k^{(1)}) + \sqrt{2\eta^{(1)}\tau^{(1)}} \boldsymbol{\xi}_k^{(1)},
\tag{14}
$$

where $\eta^{(\cdot)}$ is the learning rate, $\boldsymbol{\xi}_k^{(\cdot)}$ is a standard $d$-dimensional Gaussian noise and each subprocess follows a stochastic gradient Langevin dynamics (SGLD) (Welling & Teh, 2011). Further assuming the energy normality assumption C1 in section C.1, there exists a $\sigma_p$ such that

$$
\widetilde{U}(\boldsymbol{\beta}^{(p)}) - \widetilde{U}(\boldsymbol{\beta}^{(p+1)}) \sim \mathcal{N}(U(\boldsymbol{\beta}^{(p)}) - U(\boldsymbol{\beta}^{(p+1)}), 2\sigma_p^2), \text{ for any } \boldsymbol{\beta}^{(p)} \sim \pi^{(p)}, \tag{15}
$$

where $p \in \{1, 2, \cdots, P-1\}$, in what follows, Deng et al. (2020) proposed the bias-corrected swap function as follows

$$
a\widetilde{S}(\boldsymbol{\beta}_{k+1}^{(p)}, \boldsymbol{\beta}_{k+1}^{(p+1)}) = a \cdot \left( 1 \wedge e^{\left(\frac{1}{\tau^{(p)}} - \frac{1}{\tau^{(p+1)}}\right)\left(\widetilde{U}(\boldsymbol{\beta}_{k+1}^{(p)}) - \widetilde{U}(\boldsymbol{\beta}_{k+1}^{(p+1)}) - \left(\frac{1}{\tau^{(p)}} - \frac{1}{\tau^{(p+1)}}\right)\sigma_p^2\right)} \right), \tag{16}
$$

where $p \in \{1, 2, \cdots, P-1\}$, $\left(\frac{1}{\tau^{(p)}} - \frac{1}{\tau^{(p+1)}}\right)\sigma_p^2$ is a correction term to avoid the bias and the swap intensity $a$ can be then set to $\frac{1}{\min\{\eta^{(p)}, \eta^{(p+1)}\}}$ for convenience. Namely, given a particle pair at location $(\beta^{(p)}, \beta^{(p+1)})$ in the $k$-th iteration, the conditional probability of the swap follows that

$$
\mathbb{P}(\boldsymbol{\beta}_{k+1} = (\beta^{(p+1)}, \beta^{(p)}) | \boldsymbol{\beta}_k = (\beta^{(p)}, \beta^{(p+1)})) = \widetilde{S}(\beta^{(p)}, \beta^{(p+1)}),
$$
$$
\mathbb{P}(\boldsymbol{\beta}_{k+1} = (\beta^{(p)}, \beta^{(p+1)}) | \boldsymbol{\beta}_k = (\beta^{(p)}, \beta^{(p+1)})) = 1 - \widetilde{S}(\beta^{(p)}, \beta^{(p+1)}).
$$

## A.2 INHOMOGENOUS SWAP INTENSITY VIA DIFFERENT LEARNING RATES

Assume the high-temperature process also applies $\eta^{(p+1)} \geq \eta^{(p)} > 0$, where $p \in \{1, 2 \cdots, P-1\}$. At time $t$, the swap intensity can be interpreted as being 0 in a time interval $[t, t + \eta^{(p+1)} - \eta^{(p)})$ and being $\frac{1}{\eta^{(p)}}$ in $[t + \eta^{(p+1)} - \eta^{(p)}, t + \eta^{(p+1)})$. Since the coupled Langevin diffusion process converges to the same joint distribution regardless of the swaps and the swap intensity $a$ varies in a bounded domain, the convergence of numerical schemes is not much affected except the numerical error.

## A.3 ACCURACY-ACCELERATION TRADE-OFF

Despite the exponential acceleration potential, the average bias-corrected swap rate is significantly reduced such that $\mathbb{E}[\widetilde{S}] = O\left(Se^{-\left(\frac{1}{\tau^{(p)}} - \frac{1}{\tau^{(p+1)}}\right)^2 \frac{\sigma_p^2}{8}}\right)$ (Deng et al., 2021), where $S$ is the swap function following Eq.(2) in full-batch settings. Including more parallel chains are promising to alleviate this issue, however, it is often inevitable to sacrifice some accuracy to obtain more swaps and accelerations, as discussed in section 3.4. As such, it inspires us to devise the user-friendly SGD-based approximate exploration kernels.

## A.4 CONNECTION TO NON-CONVEX OPTIMIZATION

Parallel to our work, a similar SGLD×SGD framework was proposed by Dong & Tong (2021) for non-convex optimization, where SGLD and SGD work as exploration and exploitation kernels,

respectively. By contrast, our algorithm performs *exactly in the opposite* for uncertainty approximation because SGLD is theoretically more appealing for the exploitations based on small learning rates instead of explorations, while the widely-adopted SGDs are quite attractive in exploration due to its user-friendly nature and ability in exploring wide optima.

If we manipulate the scheme to propose *an exact swap* in each window instead of *at most one* in the current version, the algorithm shows a better potential in non-convex optimization. In particular, a larger window size corresponds to a slower decay of temperatures in simulated annealing (SAA) (Mangoubi & Vishnoi, 2018). Such a mechanism yields a larger hitting probability to move into a sub-level set with lower energies (losses) and a better chance to hit the global optima. Nevertheless, the manipulated algorithm possess the natural of parallelism in cyclical fashions.

## A.5 OTHERS

**Swap time** refers to the communication time to conduct a swap in each attempt. For example, chain pair $(p, p+1)$ of ADJ requires to wait for the completion of chain pairs $(1, 2), (2, 3), \cdots, (p-1, p)$ to attempt the swap and leads to swap time of $O(P)$; however, SEO, DEO, and DEO$_\star$ don't have this issue because in each iteration, only even or odd chain pairs are attempted to swap, hence the swap time is $O(1)$.

**Round trip time** refers to the time (stochastic variable) used in a round trip. A round trip is completed when a particle in the $p$-th chain, where $p \in [P] := \{1, 2, \cdots, P\}$, hits the index boundary index at both 1 and $P$ and returns back to its original index $p$.

## B ANALYSIS OF ROUND TRIPS

To facilitate the theoretical analysis, we follow Syed et al. (2021) and make the following assumptions

- (B1) Stationarity: Each sub-process has achieved the stationary distribution $\boldsymbol{\beta}^{(p)} \sim \pi^{(p)}$ for any $p \in [P]$;

- (B2) Weak independence: For any $\bar{\boldsymbol{\beta}}^{(p)}$ simulated from the $p$-th chain conditional on $\boldsymbol{\beta}^{(j)}$, $U(\bar{\boldsymbol{\beta}}^{(j)})$ and $U(\boldsymbol{\beta}^{(j)})$ are independent.

### B.1 ANALYSIS OF ROUND TRIP TIME

*Proof of Lemma 1.* For $t \in \mathbb{N}$, define $Z_t \in [P] = \{1, 2, \cdots, P\}$ as the index of the chain a particle arrives after $t$ windows. Define $\delta_t \in \{1, -1\}$ to indicate the direction of the swap a particle intends to make during the $t$-th window; i.e., the swap is between $Z_t$ and $Z_t + 1$ if $\delta_t = 1$ and is between $Z_t$ and $Z_t - 1$ if $\delta_t = -1$.

Define $U := \min\{t \geq 0 : Z_t = P, \delta_t = -1\}$ and $V := \min\{t \geq 0 : Z_t = 1, \delta_t = 1\}$. Define $r_p := \mathbb{P}[$ reject the swap between Chain $p$ and Chain $p+1$ for one time $]$ . Define $u_{p,\delta} := \mathbb{E}[U|Z_0 = p, \delta_0 = \delta]$ for $\delta \in \{1, -1\}$ and $v_{p,\delta} := \mathbb{E}[V|Z_0 = p, \delta_0 = \delta]$. Then the expectation of round trip time $T$ is

$$\mathbb{E}[T] = W(u_{1,1} + u_{P,-1}). \tag{17}$$

By the Markov property, for $u_{p,\delta}$, we have

$$u_{p,1} = r_p^W (u_{p,-1} + 1) + (1 - r_p^W)(u_{p+1,1} + 1) \tag{18}$$

$$u_{p,-1} = r_{p-1}^W (u_{p,1} + 1) + (1 - r_{p-1}^W)(u_{p-1,-1} + 1), \tag{19}$$

where $r_p^W$ denotes the rejection probability of a particle in a window of size $W$ at the $p$-th chain.

According to Eq.(18) and Eq.(19), we have

$$u_{p+1,1} - u_{p,1} = r_p^W (u_{p+1,1} - u_{p,-1}) - 1 \tag{20}$$

$$u_{p,-1} - u_{p-1,-1} = r_{p-1}^W (u_{p,1} - u_{p-1,-1}) + 1. \tag{21}$$

Define $\alpha_p = u_{p,1} - u_{p-1,-1}$. Then by definition, Eq.(20), and Eq.(21), we have

$$
\begin{aligned}
\alpha_{p+1} - \alpha_p &= (u_{p+1,1} - u_{p,-1}) - (u_{p,1} - u_{p-1,-1}) \\
&= (u_{p+1,1} - u_{p,1}) - (u_{p,-1} - u_{p-1,-1}) \\
&= r_p^W \alpha_{p+1} - r_{p-1}^W \alpha_p - 2,
\end{aligned}
$$

which implies that

$$
a_{p+1} - a_p = -2, \tag{22}
$$

for $a_p := (1 - r_{p-1}^W)\alpha_p$. Thus, by Eq.(22), we have

$$
a_p = a_2 - 2(p - 2). \tag{23}
$$

By definition, $u_{1,-1} = u_{1,1} + 1$. According to Eq.(20), for $p = 1$, we have

$$
\begin{aligned}
a_2 &= (1 - r_1^W)(u_{2,1} - u_{1,-1}) \\
&= (1 - r_1^W)\left[u_{1,1} - u_{1,-1} + r_1^W(u_{2,1} - u_{1,-1}) - 1\right] \\
&= -2(1 - r_1^W) + r_1^W a_2.
\end{aligned} \tag{24}
$$

Since $r_p \in (0, 1)$ for $1 \le p \le P$, Eq.(24) implies $a_2 = -2$ which together with Eq.(23) implies

$$
(1 - r_{p-1}^W)\alpha_p = a_p = -2(p - 1), \tag{25}
$$

and therefore

$$
r_{p-1}^W \alpha_p = -2(p - 1)\frac{r_{p-1}^W}{1 - r_{p-1}^W}. \tag{26}
$$

According to Eq.(21) and Eq.(26), we have

$$
u_{P,1} - u_{1,1} = \sum_{p=1}^{P-1} r_p^W(u_{p+1,1} - u_{p,-1}) - (P - 1) \tag{27}
$$

$$
= \sum_{p=1}^{P-1} r_p^W \alpha_{p+1} - (P - 1) \tag{28}
$$

$$
= -2\sum_{p=1}^{P-1} \frac{r_p^W}{1 - r_p^W} p - (P - 1). \tag{29}
$$

Since $u_{P,1} = 1$, we have

$$
\begin{aligned}
u_{1,1} &= u_{P,1} + 2\sum_{p=1}^{P-1} \frac{r_p^W}{1 - r_p^W} p + (P - 1) \\
&= P + 2\sum_{p=1}^{P-1} \frac{r_p^W}{1 - r_p^W} p.
\end{aligned} \tag{30}
$$

Similarly, for $v_{p,\delta}$, we also have

$$
\begin{aligned}
v_{p,1} &= r_p^W(v_{p,-1} + 1) + (1 - r_p^W)(v_{p+1,1} + 1) \\
v_{p,-1} &= r_{p-1}^W(v_{p,1} + 1) + (1 - r_{p-1}^W)(v_{p-1,-1} + 1).
\end{aligned}
$$

With the same analysis, we have

$$
b_{p+1} - b_p = -2 \tag{31}
$$

$$
v_{p,-1} - v_{p-1,-1} = r_{p-1}^W(v_{p,1} - v_{p-1,-1}) + 1, \tag{32}
$$

where $b_p := (1 - r_{p-1}^W)\beta_p$ and $\beta_p := v_{p,1} - v_{p-1,-1}$. According to Eq.(32), we have

$$
v_{P,-1} - v_{1,-1} = \sum_{p=1}^{P-1} r_p^W \beta_{p+1} + (P - 1). \tag{33}
$$

By definition, $v_{P,1} = v_{P,-1} + 1$. According to Eq.(32), for $p = P$, we have

$$
\begin{aligned}
b_P &= (1 - r_{P-1}^W)(v_{P,1} - v_{P-1,-1}) \\
&= (1 - r_{P-1}^W)\left[v_{P,1} - v_{P,-1} + r_{P-1}^W(v_{P,1} - v_{P-1,-1}) + 1\right] \\
&= r_{P-1}^W b_P + 2(1 - r_{P-1}^W).
\end{aligned}
\tag{34}
$$

Since $r_p \in (0,1)$ for $1 \le p \le P$, Eq.(34) implies $b_P = 2$. Then according to Eq.(31), we have

$$
b_p = 2(P - p + 1),
$$

and therefore

$$
r_{p-1}^W \beta_p = 2 \frac{r_{p-1}^W}{1 - r_{p-1}^W}(P - p + 1).
\tag{35}
$$

By Eq.(33) and Eq.(35), we have

$$
v_{P,-1} - v_{1,-1} = 2 \sum_{p=1}^{P-1} \frac{r_p^W}{1 - r_p^W}(P - p) + (P - 1).
$$

Since $v_{1,-1} = 1$, we have

$$
v_{P,-1} = 2 \sum_{p=1}^{P-1} \frac{r_p^W}{1 - r_p^W}(P - p) + P.
\tag{36}
$$

According to Eq.(17), Eq.(30) and Eq.(36), we have

$$
\mathbb{E}[T] = W(u_{1,1} + u_{P,-1}) = 2WP + 2WP \sum_{p=1}^{P-1} \frac{r_p^W}{1 - r_p^W}.
$$

$\square$

The proof is a generalization of Theorem 1 (Syed et al., 2021); when $W = 1$, the generalized DEO scheme recovers the DEO scheme. For the self-consistency of our analysis, we present it here anyway.

## B.2 ANALYSIS OF OPTIMAL WINDOW SIZE

*Proof of Theorem 1.* By treating $W \ge 1$ as a continuous variable and taking the derivative of $\mathbb{E}[T]$ with respect to $W$ and we can get

$$
\begin{aligned}
\frac{\partial}{\partial W}\mathbb{E}[T] &= 2P\left[1 + \sum_{p=1}^{P-1} \frac{r_p^W}{(1 - r_p^W)} + W \sum_{p=1}^{P-1} \frac{r_p^W \log r_p}{(1 - r_p^W)^2}\right] \\
&= 2P\left\{1 + \sum_{p=1}^{P-1} \frac{r_p^W - r_p^{2W} + W r_p^W \log r_p}{(1 - r_p^W)^2}\right\} \\
&= 2P\left\{1 + \sum_{p=1}^{P-1} r_p^W \frac{1 - r_p^W + W \log r_p}{(1 - r_p^W)^2}\right\}.
\end{aligned}
\tag{37}
$$

Assume that $r_p = r \in (0,1)$ for $1 \le p \le P$. Then we have

$$
\mathbb{E}[T] = 2WP + 2WP(P - 1)\frac{r^W}{1 - r^W}.
\tag{38}
$$

$$
\frac{\partial}{\partial W}\mathbb{E}[T] = \frac{2P}{(1 - r^W)^2}\left\{(1 - r^W)^2 + (P - 1)r^W(1 - r^W + W \log r)\right\}.
\tag{39}
$$

Define $x := r^W \in (0,1)$. Hence $W = \log_r(x) = \frac{\log x}{\log r}$ and

$$
\frac{\partial}{\partial W}\mathbb{E}[T] = f(x) := \frac{2P}{(1 - x)^2}\left\{(1 - x)^2 + (P - 1)x(1 - x + \log x)\right\}
\tag{40}
$$

Thus it suffices to analyze the sign of the function $g(x) := (1-x)^2 + (P-1)x(1-x+\log(x))$ for $x \in (0,1)$. For $g(x)$, we have

$$g'(x) = (4-2P)(x-1) + (P-1)\log(x)$$

and

$$g''(x) = 4 - 2P + \frac{P-1}{x}.$$

Thus, $\lim_{x \to 0^+} g'(x) = -\infty$, $\lim_{x \to 1} g'(x) = g'(1) = 0$ and $g''(x)$ is monotonically decreasing for $x > 0$.

1. For $P > 2$, we know that $g''(x) > 0$ when $0 < x < \frac{P-1}{2(P-2)}$ and $g''(x) < 0$ when $x < \frac{P-1}{2(P-2)}$. Therefore, $g'(x)$ is maximized at $\frac{P-1}{2(P-2)}$.

   (a) If $P = 3$, by $\log(1+y) < y$ for $y > 0$, we have $g'(x) = 2\big(\log(1+(x-1))-(x-1)\big) < 0$ for any $x \in (0,1)$. Thus $g(x) > g(1) = 0$ for $x \in (0,1)$ and therefore $\frac{\partial}{\partial W}\mathbb{E}[T] > 0$ for $W \in \mathbb{N}^+$. $\mathbb{E}[T]$ is globally minimized at $W = 1$.

   (b) If $P > 3$, we have the following lemma and the proof is postponed in section B.2.1.
   **Lemma 3** (Uniqueness of the solution). *For $P > 3$, there exists a unique solution $x^* \in (0,1)$ such that $g(x) > 0$ for $\forall x \in (0, x^*)$ and $g(x) < 0$ for $\forall x \in (x^*, 1)$. Moreover, $W^* = \log_r(x^*)$ is the globally minimizer for the round trip time.*

2. For $P = 2$, we know that $g''(x) > 0$ for $x \in (0,1)$. Thus $g'(x) < g(1) = 0$ for $x \in (0,1)$ and therefore $g(x) > g'(1) = 0$ for $x \in (0,1)$. Then according to Eq.(37), $\frac{\partial}{\partial W}\mathbb{E}[T] > 0$ for $W \in \mathbb{N}^+$. $\mathbb{E}[T]$ is globally minimized at $W = 1$.

In what follows, we proceed to prove that $\frac{1}{P \log P}$ is a good approximation to $x^*$. In fact, we have

$$g\left(\frac{1}{P \log P}\right) = \left(1 - \frac{1}{P \log P}\right)^2 + \frac{P-1}{P \log P}\left(1 - \frac{1}{P \log P} - \log P - \log(\log P)\right)$$

$$= 1 + o\left(\frac{1}{P}\right) - 1 + \frac{1}{\log P} - \frac{\log(\log P)}{\log P} + O\left(\frac{1}{P}\right)$$

$$= -\frac{\log(\log P)}{\log P} + O\left(\frac{1}{\log P}\right)$$

Thus, $\lim_{P \to \infty} g(\frac{1}{P \log P}) = 0$.

For $x = \frac{1}{P \log P}$, we have $W = \log_r\left(\frac{1}{P \log P}\right) = \frac{\log P + \log \log P}{-\log r}$. Then according to Eq.(38), for $P \geq 4$,

$$\mathbb{E}[T] = 2P\frac{\log P + \log \log P}{-\log r}\left[1 + (P-1)\frac{\frac{1}{P \log P}}{1 - \frac{1}{P \log P}}\right]$$

$$= \left(1 + \frac{1}{\log P}\frac{1}{1 - \frac{1}{P \log P}}\right)\frac{2}{-\log r}(P \log P + P \log \log P)$$

$$\leq \left(1 + \frac{1}{\log 4}\frac{1}{1 - \frac{1}{4 \log 4}}\right)\frac{2}{-\log r}(P \log P + P \log \log P) \qquad (41)$$

$$< \frac{4}{-\log r}(P \log P + P \log \log P)$$

In conclusion, for $P = 2, 3$, the maximum round trip rate is achieved when the window size $W = 1$. For $P \geq 4$, with the window size $W = \frac{\log P + \log \log P}{-\log r}$, the round trip rate is at least $\Omega\left(\frac{-\log r}{\log P}\right)$. $\quad\square$

**Remark:** Given finite chains with a large rejection rate, the round trip time is only of order $O(P \log P)$ by setting the optimal window size $W \approx \left\lceil \frac{\log P + \log \log P}{-\log r} \right\rceil$. By contrast, the vanilla DEO

scheme with a window of size 1 yields a much longer time of $O(P^2)$, where $\frac{1}{-\log(r)} = O(\frac{r}{1-r})$ based on Taylor expansion and a large $r \gg 0$.

### B.2.1 TECHNICAL LEMMA

*Proof of Lemma 3.* To help illustrate the analysis below, we plot the graphs of $g'(x)$ and $g(x)$ for $x \in (0,1)$ and $P = 5$ in Figure 9. For $P > 3$, since $g'(x)$ is maximized at $x = \frac{P-1}{2(P-2)} \in (0,1)$ with $g''(x) > 0$ when $0 < x < \frac{P-1}{2(P-2)}$ and $g''(x) < 0$ when $\frac{P-1}{2(P-2)} < x < 1$, $\lim_{x \to 0^+} g'(x) = -\infty$, and $g'(1) = 0$, we know that $g'(\frac{P-1}{2(P-2)}) > 0$ and there exists $x_0 \in (0, \frac{P-1}{2(P-2)})$ such that $g'(x) < 0$ (i.e., $g'(x)$ is monotonically decreasing) for any $x \in (0, x_0)$ and $g'(x) > 0$ (i.e., $g'(x)$ is monotonically increasing) for any $x \in (x_0, 1)$. Then $g(x)$ on $(0,1)$ is globally minimized at $x = x_0$. Moreover, $\lim_{x \to 0^+} g(x) = 1$ and $g(1) = 0$. Thus, $g(x_0) < g(1) = 0$ and there exists $x^* \in (0, x_0) \subsetneq (0,1)$ such that $g(x) > 0$ if $x \in (0, x^*)$ and $g(x) < 0$ if $x \in (x^*, 1)$.

Meanwhile, by Eq.(37) and the definition of $x$, we know that the sign of $\frac{\partial}{\partial W}\mathbb{E}[T]$ is the same with that of $g(x)$ for $W = \log_r(x)$. Thus, $\frac{\partial}{\partial W}\mathbb{E}[T] < 0$ when $W < W^* := \log_r(x^*)$ and $\frac{\partial}{\partial W}\mathbb{E}[T] > 0$ when $W > W^*$, which implies that $\mathbb{E}[T]$ is globally minimized at $W^* = \log_r(x^*)$ with some $x^* \in (0,1)$.

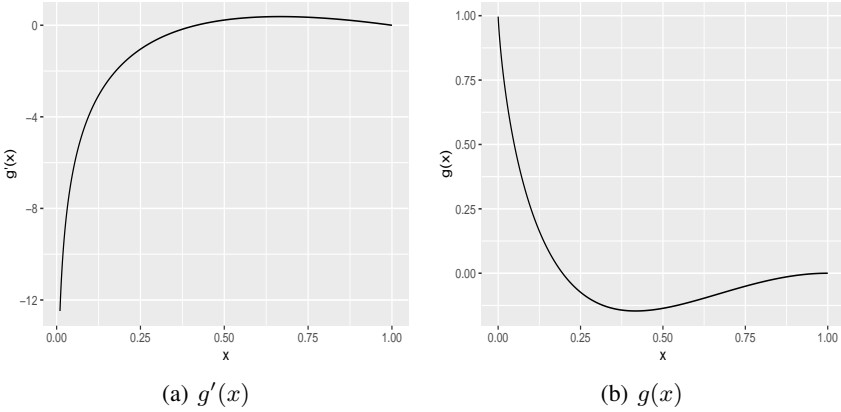

(a) $g'(x)$           (b) $g(x)$

Figure 9: Illustration of functions $g'(x)$ and $g(x)$ for $x \in (0,1)$ and $P = 5$

$\square$

### B.3 DISCUSSIONS ON THE OPTIMAL NUMBER OF CHAINS IN BIG DATA

To shed light on the suggestions of the optimal number of chains, we lay out two additional assumptions:

- (B3) Integrability: $U^3$ is integrable with respect to $\pi^{(1)}$ and $\pi^{(P)}$.

- (B4) Approximate geometric spacing: Assume that $\frac{\tau^{(p+1)}}{\tau^{(p)}} \equiv \left(\frac{\tau^{(P)}}{\tau^{(1)}}\right)^{\frac{1}{P-1}} + o(\frac{1}{P})$ for $p \in [P-1]$.

Assumption B3 is a standard assumption in Syed et al. (2021) to establish the convergence of the summation of the rejection rates. Assumption B4 allows extra perturbations to the geometric spacing assumption (Kofke, 2002) and is empirically verified in the CIFAR100 example in Figure 7(c).

*Proof of corollary 1.* Before we start the proof, we denote by $\tilde{r}$ and $\tilde{s}$ the average rejection and acceptance rates in full-batch settings, respectively. According to section 5.1 in Syed et al. (2021), Assumptions B1, B2 and B3 lead to

$$\tilde{r} = \frac{\Lambda}{P} + O\left(\frac{1}{P^3}\right), \tag{42}$$

where $\Lambda$ is the barrier, which becomes larger if $\pi^{(1)}$ and $\pi^{(P)}$ have a smaller probability overlap (Predescu et al., 2004; Syed et al., 2021). Then acceptance rate in full-batch settings is

$$\tilde{s} = 1 - \tilde{r} = 1 - \frac{\Lambda}{P} + O\left(\frac{1}{P^3}\right). \tag{43}$$

By Lemma D4 (Deng et al., 2021) and Eq.(15), a noisy energy estimator with variance $\frac{(\tau^{(p)} - \tau^{(p+1)})^2}{\tau^{(p)2}\tau^{(p+1)2}}\sigma_p^2$ yields an average swap rate of $O\left(\tilde{s}\exp\left(-\frac{(\tau^{(p)} - \tau^{(p+1)})^2}{8\tau^{(p)2}\tau^{(p+1)2}}\right)\sigma_p^2\right)$. In what follows, the average rejection rate in mini-batch settings becomes

$$r = 1 - O\left(\tilde{s}\exp\left(-\frac{(\tau^{(p)} - \tau^{(p+1)})^2}{8\tau^{(p)2}\tau^{(p+1)2}}\right)\sigma_p^2\right), \tag{44}$$

where $p \in [P]$ and more accurate estimates is studied in proposition 2.4 (Gelman et al., 1997).

Define $\Delta_\tau = \frac{\tau^{(P)}}{\tau^{(1)}}$. By assumption B4 and Taylor's theorem, we have

$$\left(\frac{\tau^{(p+1)} - \tau^{(p)}}{\tau^{(p)}}\right)^2 := \left(\Delta_\tau^{\frac{1}{P-1}} - 1 + o\left(\frac{1}{P}\right)\right)^2 = \left(\frac{\log \Delta_\tau}{P-1}\right)^2 + o\left(\frac{1}{P^2}\right). \tag{45}$$

Denote $\gamma_p = \frac{\sigma_p}{\tau^{(p+1)}}$. It follows that

$$\exp\left(-\frac{(\tau^{(p)} - \tau^{(p+1)})^2}{8\tau^{(p)2}\tau^{(p+1)2}}\sigma_p^2\right)$$
$$= \exp\left(-\left(\frac{(\log \Delta_\tau)^2}{8(P-1)^2} + o\left(\frac{1}{P^2}\right)\right)\frac{\sigma_p^2}{\tau^{(p)2}}\right) \tag{46}$$
$$= \exp\left(-\frac{(\gamma_p \log \Delta_\tau)^2}{8(P-1)^2} + o\left(\frac{1}{P^2}\right)\right).$$

Plugging it to Eq.(44), we have

$$r = 1 - O\left(\left(1 - \frac{\Lambda}{P} + O\left(\frac{1}{P^3}\right)\right)\exp\left(-\frac{(\gamma_p \log \Delta_\tau)^2}{8(P-1)^2} + o\left(\frac{1}{P^2}\right)\right)\right). \tag{47}$$

Recall in Eq.(41), the round trip time follows that

$$\mathbb{E}[T] = O\left(\frac{P \log P}{-\log r}\right). \tag{48}$$

By $\log(1-t) \le -t$ for $t \in [0,1)$ and $\frac{1}{1-t} = 1 + t + O(t^2)$ for a small $t$, we have

$$-\frac{1}{\log r} = -\frac{1}{\log\left\{1 - O\left(\left(1 - \frac{\Lambda}{P} + O\left(\frac{1}{P^3}\right)\right)\exp\left(-\frac{(\gamma_p \log \Delta_\tau)^2}{8(P-1)^2} + o\left(\frac{1}{P^2}\right)\right)\right)\right\}}$$
$$\le \frac{1}{O\left(\left(1 - \frac{\Lambda}{P} + O\left(\frac{1}{P^3}\right)\right)\exp\left(-\frac{(\gamma_p \log \Delta_\tau)^2}{8(P-1)^2} + o\left(\frac{1}{P^2}\right)\right)\right)} \tag{49}$$
$$= O\left(\left(1 + \frac{\Lambda}{P} + O\left(\frac{1}{P^3}\right)\right)\exp\left(\frac{(\gamma_p \log \Delta_\tau)^2}{8(P-1)^2} + o\left(\frac{1}{P^2}\right)\right)\right).$$

Combining Eq.(49) and Eq.(48), we have

$$\mathbb{E}[T] = O\left(P \log P\left(1 + \frac{\Lambda}{P} + O\left(\frac{1}{P^3}\right)\right)\exp\left(\frac{\mu_p^2}{(P-1)^2} + O\left(\frac{1}{P^2}\right)\right)\right)$$
$$= O\left(P \log P \exp\left(\frac{\mu_p^2}{P^2}\right)\right), \tag{50}$$

where $\mu_p := \frac{\gamma_p \log \Delta_\tau}{2\sqrt{2}}$. The optimal $P$ to minimize $\mathbb{E}[T] = O\left(P \log P \exp\left(\frac{\mu_p^2}{P^2}\right)\right)$ is equivalent to the minimizer of $h_p(x) = \log\left\{x \log x \exp\left(\frac{\mu_p^2}{x^2}\right)\right\} = \log x + \log \log x + \frac{\mu_p^2}{x^2}$ for $x \geq 4$. Then

$$h_p'(x) = \frac{1}{x} + \frac{1}{x \log x} - \frac{2\mu_p^2}{x^3} = \frac{x^2(1 + \frac{1}{\log x}) - 2\mu_p^2}{x^3}, \tag{51}$$

Define $s(x) = x^2(1 + \frac{1}{\log x})$. Then we have

$$s'(x) = 2x + \frac{2x \log x - x}{(\log x)^2} > 0, \tag{52}$$

for $x \geq 4$. Thus $s(x)$ increases with $x$ for $x \geq 4$. Obviously, $\lim_{x \to \infty} s(x) = \infty$. Since $\mu_p \gg 1$, we know $s(4) < 0$. Thus, there is a single zero point $P_\star \in (4, \infty)$ of $s(x)$ such that $h_p'(x) < 0$ if $x \in [4, P_\star)$ and $h_p'(x) > 0$ if $x > P_\star$. Therefore, $h_p(x)$ is minimized at $P_\star$ with

$$P_\star^2\left(1 + \frac{1}{\log P_\star}\right) \in \left(2 \min_p \mu_p^2, 2 \max_p \mu_p^2\right).$$

For any $P_\star \geq 2$, we have $\sqrt{2/(1 + 1/\log P_\star)} \in (1.1, \sqrt{2})$. Thus, we have the optimal number of chains in mini-batch settings following that

$$P_\star \in (1.1 \min_p \mu_p, \sqrt{2} \max_p \mu_p) \in \left(\min_p \frac{\sigma_p}{3\tau^{(p+1)}} \log \Delta_\tau, \max_p \frac{\sigma_p}{2\tau^{(p+1)}} \log \Delta_\tau\right). \tag{53}$$

$\square$

### B.4 EMPIRICAL JUSTIFICATION OF THE OPTIMAL NUMBER OF CHAINS

To obtain an estimate of the optimal number of chains for PT in big data problems, we study the standard deviation of the noisy energy estimators on CIFAR100 via ResNet20 models. We first pre-train a model and then try different learning rates to check the corresponding standard deviation of energy estimators. The reason we abandon the temperature variable is that the widely used data augmentation has drastically affected the estimation of the temperature (Wenzel et al., 2020). Motivated by the linear relation between the learning rate and the temperature in Eq.(6) and the cold posterior affect with the temperature much smaller than 1 (Aitchison, 2021), applying Figure 10 and Eq.(53) concludes that $P_\star$ is *achieved in an order of thousands in the CIFAR100 example.*

The conclusion is different from Syed et al. (2021) since big data problems require a much smaller swap rate to maintain the unbiasedness of the swaps. On the one hand, insufficient chains may lead to insignificant swap rates to generate effective accelerations, but on the other hand, introducing too many chains may be too cost in terms of limited memory and computational budget.

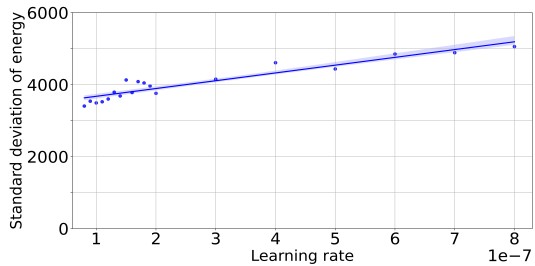

Figure 10: Analysis of standard deviation of energy estimators with respect to different learning rates through ResNet20 on CIFAR100 dataset. Note that we have transformed the average likelihood into the sum of likelihood with a total number of 50,000 datapoints. Hence, the learning rate is also reduced by 50,000 times.

# C APPROXIMATION ERROR FOR SGD-BASED EXPLORATION KERNELS

This section proposes to analyze the approximation error when we adopt SGDs as the approximate exploration kernels. Since it is independent of section B, different assumptions may be required.

The first subsection shows that for any Gaussian-distributed energy difference estimator $\widetilde{U}(\boldsymbol{\beta}^{(p)}) - \widetilde{U}(\boldsymbol{\beta}^{(p+1)})$, there exists an optimal $\mathbb{C}_\star$ such that the *deterministic swap condition* $\widetilde{U}(\boldsymbol{\beta}^{(p+1)}) + \mathbb{C}_\star < \widetilde{U}(\boldsymbol{\beta}^{(p)})$ perfectly approximates the random event $\widetilde{S}(\boldsymbol{\beta}^{(p)}, \boldsymbol{\beta}^{(p+1)}) > u$, where $u \sim \text{Unif}[0, 1]$.

## C.1 GAP BETWEEN ACCEPTANCE RATES

To approximate the error when we adopt the *deterministic swap condition* Eq.(8), we first assume the following condition

- (C1) Energy normality: The stochastic energy estimator for each chain follows a normal distribution with a fixed variance.

The above assumption has been widely assumed by Ceperley & Dewing (1999); Bardenet et al. (2017); Seita et al. (2017), which can be easily extended to the asymptotic normality assumption depending on large enough batch sizes (Quiroz et al., 2019; Deng et al., 2021).

*Proof of Lemma 2.* Denote $\partial T_p = \frac{1}{\tau^{(p)}} - \frac{1}{\tau^{(p+1)}} > 0$ for any $p \in [P-1]$, the energy difference $\partial U_p = U(\boldsymbol{\beta}^{(p)}) - U(\boldsymbol{\beta}^{(p+1)})$. By the energy normality assumption C1 and Eq.(15), the energy difference estimator follows that $\partial \widetilde{U}_p = \partial U_p(\cdot) + \sqrt{2}\sigma_p \xi$, where $\xi \sim \mathcal{N}(0, 1)$.

Recall from Eq.(16) that the bias-corrected acceptance rate follows that

$$
\begin{aligned}
\widetilde{S}(\boldsymbol{\beta}_t^{(p)}, \boldsymbol{\beta}_t^{(p+1)}) &= 1 \wedge e^{\partial T_p\left(\widetilde{U}(\boldsymbol{\beta}_t^{(p)}) - \widetilde{U}(\boldsymbol{\beta}_t^{(p+1)})\right) - \partial T_p^2 \sigma_p^2} \\
&= 1 \wedge e^{\partial T_p \partial U_p + \sqrt{2}\partial T_p \sigma_p \xi - \partial T_p^2 \sigma_p^2},
\end{aligned}
\tag{54}
$$

(i) For a uniformly distributed variable $\mu \sim \text{Unif}[0, 1]$, the base swap function satisfies that

$$
\begin{aligned}
\mathbb{P}\left(\widetilde{S}(\boldsymbol{\beta}_t^{(p)}, \boldsymbol{\beta}_t^{(p+1)}) > \mu | \partial U_p\right) &= \int_0^1 \mathbb{P}(\widetilde{S}(\boldsymbol{\beta}_t^{(p)}, \boldsymbol{\beta}_t^{(p+1)}) > u | \partial U_p) du \\
&= \int_0^1 \mathbb{P}(e^{\partial T_p \partial U_p + \sqrt{2}\partial T_p \sigma_p \xi - \partial T_p^2 \sigma_p^2} > u | \partial U_p) du \\
&= \int_0^1 \mathbb{P}\left(\xi > \frac{-\partial U_p + \partial T_p \sigma_p^2 + \frac{\log u}{\partial T_p}}{\sqrt{2}\sigma_p}\right) du
\end{aligned}
\tag{55}
$$

(ii) For the deterministic swaps based on Eq.(8), the approximate swap function follows that

$$
\mathbb{P}\left(\widetilde{U}(\boldsymbol{\beta}_k^{(p+1)}) + \mathbb{C} < \widetilde{U}(\boldsymbol{\beta}_k^{(p)}) | \partial U_p\right) = \mathbb{P}(\mathbb{C} < \partial U_p + \sqrt{2}\sigma_p \xi | \partial U_p) = \mathbb{P}\left(\xi > \frac{-\partial U_p + \mathbb{C}}{\sqrt{2}\sigma_p}\right).
\tag{56}
$$

Denote by $D(u, \mathbb{C}) = \left| \mathbb{P}\left( \xi > \frac{-\partial U_p + \partial T_p \sigma_p^2 + \frac{\log u}{\partial T_p}}{\sqrt{2}\sigma_p} \right) - \mathbb{P}\left( \xi > \frac{-\partial U_p + \mathbb{C}}{\sqrt{2}\sigma_p} \right) \right| \in [0, 2)$. Combining (55) and (56), we can easily derive that

$$
\begin{aligned}
\Delta(\mathbb{C}) &= \mathbb{P}\left( \widetilde{S}(\boldsymbol{\beta}_t^{(p)}, \boldsymbol{\beta}_t^{(p+1)}) > \mu | \partial U_p \right) - \mathbb{P}\left( \widetilde{U}(\boldsymbol{\beta}_k^{(p+1)}) + \mathbb{C} < \widetilde{U}(\boldsymbol{\beta}_k^{(p)}) | \partial U_p \right) \\
&= \int_0^{e^{-\partial T_p^2 \sigma_p^2 + \partial T_p \mathbb{C}}} \underbrace{\mathbb{P}\left( \xi > \frac{-\partial U_p + \partial T_p \sigma_p^2 + \frac{\log u}{\partial T_p}}{\sqrt{2}\sigma_p} \right) - \mathbb{P}\left( \xi > \frac{-\partial U_p + \mathbb{C}}{\sqrt{2}\sigma_p} \right)}_{D(u, \mathbb{C}) \text{ for small enough } u} du \\
&\quad + \int_{e^{-\partial T_p^2 \sigma_p^2 + \partial T_p \mathbb{C}}}^1 \underbrace{\mathbb{P}\left( \xi > \frac{-\partial U_p + \partial T_p \sigma_p^2 + \frac{\log u}{\partial T_p}}{\sqrt{2}\sigma_p} \right) - \mathbb{P}\left( \xi > \frac{-\partial U_p + \mathbb{C}}{\sqrt{2}\sigma_p} \right)}_{-D(u, \mathbb{C}) \text{ for large enough } u} du \\
&= \int_0^{e^{-\partial T_p^2 \sigma_p^2 + \partial T_p \mathbb{C}}} D(u, \mathbb{C}) du - \int_{e^{-\partial T_p^2 \sigma_p^2 + \partial T_p \mathbb{C}}}^1 D(u, \mathbb{C}) du
\end{aligned}
$$

Clearly, $\Delta(\mathbb{C})$ is continuous in $\mathbb{C}$ and it is straightforward to verify that

$$
\begin{aligned}
\Delta(\partial T_p \sigma_p^2) &= \int_0^{e^{-\partial T_p^2 \sigma_p^2 + \partial T_p^2 \sigma_p^2}} D(u, \partial T_p \sigma_p^2) du - \int_{e^{-\partial T_p^2 \sigma_p^2 + \partial T_p^2 \sigma_p^2}}^1 D(u, \partial T_p \sigma_p^2) du \\
&= \int_0^1 D(u, \partial T_p \sigma_p^2) du \geq 0
\end{aligned}
$$

Similarly, $\Delta(-\infty) = -\int_0^1 D(u, \mathbb{C}) du \leq 0$. Moreover, the physical construction suggests that the threshold $\mathbb{C}$ should be strictly positive to avoid radical swap attempts, which implies that there is an optimal solution $\mathbb{C}_\star \in (0, \partial T_p \sigma_p^2]$ that solves $\Delta(\mathbb{C}_\star) = 0$.

$\square$

**Remark 1.** *Note that the optimal $\mathbb{C}_\star$ may lead to few swaps given a finite number of iterations and sometimes it is suggested to trade in some accuracy to obtain more accelerations (Deng et al., 2020). As such, by setting a desired swap rate via the iterate (13), we can estimate the unknown threshold $\mathbb{C}$ by stochastic approximation (Robbins & Monro, 1951). Furthermore, as discussed in Lemma B2 (Deng et al., 2021), the average swap rate follows that $\mathbb{E}[\widetilde{S}(\boldsymbol{\beta}_t^{(p)}, \boldsymbol{\beta}_t^{(p+1)})] = O(e^{-\frac{\partial T_p^2 \sigma_p^2}{8}})$. This suggests that for a well-approximated threshold $\mathbb{C}$, the error is at most $\mathbb{E}[\Delta(\mathbb{C})] = O(e^{-\frac{\partial T_p^2 \sigma_p^2}{8}})$.*

## C.2 APPROXIMATE UPPER BOUND

The second subsection completes the proof regarding the numerical approximation of SGD-based exploration kernels. Nevertheless, we still require some standard assumptions.

- (C2) Smoothness: The function $U(\cdot)$ is $C$-smooth if there exists a positive Lipschitz constant $C$ such that $\|\nabla U(x) - \nabla U(y)\|_2 \leq C\|x - y\|_2$ for every $x, y \in \mathbb{R}^d$.

- (C3) Dissipativity: The function $U(\cdot)$ is $(a, b)$-dissipative if there exist positive constants $a$ and $b$ such that $\langle x, \nabla U(x) \rangle \geq a\|x\|^2 - b$.

- (C4) Gradient normality: The stochastic gradient noise $\varepsilon(\cdot)$ in Eq.(6) follows a Normal distribution $\mathcal{N}(0, \mathcal{M})$ with a fixed positive definite matrix $\mathcal{M}$.

The assumptions C2 and C3 are standard to show the geometric ergodicity of Langevin diffusion and the diffusion approximations for non-convex functions (Mattingly et al., 2002; Raginsky et al., 2017; Xu et al., 2018). The assumption C4 is directly motivated by Mandt et al. (2017) to track the Ornstein Uhlenbeck (OU) process with a Gaussian invariant measure.

*Proof of Theorem 2.* The transition kernel $\mathcal{T}$ of the continuous-time replica exchange Langevin diffusion (reLD) follows that $\mathcal{T} = \prod_p \mathcal{T}^{(p)}$, where each sub-kernel $\mathcal{T}^{(p)}$ draws a candidate $\boldsymbol{\theta} \in \mathbb{R}^d$ with the following probability

$$\mathcal{T}^{(p)}(\boldsymbol{\theta}|\boldsymbol{\beta}^{(p)}, \boldsymbol{\beta}^{(q)}) = (1 - S(\boldsymbol{\beta}_t^{(p)}, \boldsymbol{\beta}_t^{(q)}))Q_p(\boldsymbol{\theta}|\boldsymbol{\beta}^{(p)}) + S(\boldsymbol{\beta}_t^{(p)}, \boldsymbol{\beta}_t^{(q)})Q_q(\boldsymbol{\theta}|\boldsymbol{\beta}^{(q)}),$$

where $p, q \in \{1, 2, \cdots, P\}$, $|p - q| = 1$ and $q$ is selected based on the $\text{DEO}_\star$ scheme (3), $Q_p(\boldsymbol{\theta}|\boldsymbol{\beta})$ is the proposal distribution in the $p$-th chain that models the probability to draw the parameter $\boldsymbol{\theta}$ via Langevin diffusion conditioned on $\boldsymbol{\beta}$, and $S(\boldsymbol{\beta}_t^{(p)}, \boldsymbol{\beta}_t^{(q)})$ follows the swap function in Eq.(2).

Now we consider the following approximations:

$$\text{reLD} \overset{\text{I}}{\Rightarrow} \text{reSGLD} \overset{\text{II}}{\Rightarrow} \widehat{\text{DEO}_\star}\text{-SGLD} \overset{\text{III}}{\Rightarrow} \text{DEO}_\star\text{-SGLD} \overset{\text{IV}}{\Rightarrow} \text{DEO}_\star\text{-SGD},$$

where $\widehat{\text{DEO}_\star}$-SGLD resembles the $\text{DEO}_\star$ scheme except that there are *no restrictions on conducting at most one swap*.

**I: Numerical approximation via SGLD**  By Lemma 5, we show that approximating replica exchange Langevin diffusion via reSGLD yields a weak error only depending on the learning rate; similar results depending on the learning rate and the noise in gradient and energy noise have been studied in Lemma 1 of (Deng et al., 2020).

**II: Restrictions on at most one swap**  Adopting at most one swap in the $\text{DEO}_\star$ scheme includes **a stopping time**, which may affect the underlying distribution. Fortunately, the bias can be controlled and becomes smaller in big data problems, which is detailed as follows.

Note that $\widehat{\text{DEO}_\star}$-SGLD differs from $\text{DEO}_\star$ in that $\text{DEO}_\star$-SGLD freezes at most $W - 1$ attempts of swaps in each neighboring chains in each window, while $\widehat{\text{DEO}_\star}$-SGLD keeps all of these positions open. Recall that the bias-corrected swap rate follows that $\widetilde{S} = O(Se^{-O(\sigma^2)})$ by invoking Eq.(16). Following a similar technique in analyzing the bias through the difference of acceptance rates in Lemma 2, we can show that the bias is at most $\widetilde{S} - \underbrace{\mathbb{P}(\text{Acceptance rate given frozen swaps})}_{=0} =$

$O(e^{-O(\sigma^2)})$. In other words, the **restrictions on at most one swap only lead to a mild bias in big data problems** and is less of a concern compared to the local trap problems. Empirically, we have verified the relation between the bias and the variance of noisy energy estimator in a simulated example in section C.2.1.

**III: Deterministic swap condition**  By Lemma 2 and the mean-value theorem, there exists an optimal correction buffer $\mathbb{C}_\star$ to approximate the random event $\widetilde{S}(\boldsymbol{\beta}_t^{(p)}, \boldsymbol{\beta}_t^{(p+1)}) > \mu$ through $\widetilde{U}(\boldsymbol{\beta}_k^{(p)}) + \mathbb{C}_\star < \widetilde{U}(\boldsymbol{\beta}_k^{(p+1)})$ in the average sense for any $\boldsymbol{\beta}_t^{(p)}$ and $\boldsymbol{\beta}_t^{(p+1)}$.

**IV: Laplace approximation via SGD**  Assumption C4 guarantees that there exists an optimal learning rate $\eta$ to conduct Laplace approximation for the target posterior (Mandt et al., 2017). Further, by the Bernstein-von Mises Theorem, the approximation error goes to 0 as the number of total data points goes to infinity.

Combining the above approximations, there exists a finite constant $\Delta_{\max} \geq 0$ depending on the learning rates $\eta$, the variance of noisy energy estimators, and the choice of correction terms such that the total approximation error of the SGD exploration kernels with *deterministic swap condition* is upper bounded by $\Delta_{\max}$. For any joint probability density $\mu$, the distance between the distributions generated by one step of the exact transition kernel $\mathcal{T}$ and the approximate transition kernel $\mathcal{T}_\eta$ in

Eq.(7) is upper bounded by

$$\int_{\boldsymbol{\beta}} d\Omega(\boldsymbol{\beta}) \Big| \mu\mathcal{T}(\boldsymbol{\beta}) - \mu\mathcal{T}_\eta(\boldsymbol{\beta}) \Big|$$

$$\leq \int_{\boldsymbol{\beta}} d\Omega(\boldsymbol{\beta}) \Big| \int_{\boldsymbol{\phi},\boldsymbol{\psi}} d\mu(\boldsymbol{\phi},\boldsymbol{\psi}) \Delta_{\max} \Big( Q(\boldsymbol{\beta}|\boldsymbol{\phi}) - Q(\boldsymbol{\beta}|\boldsymbol{\psi}) \Big) \Big|$$

$$\leq \Delta_{\max} \int d\Omega(\boldsymbol{\beta}) \int_{\boldsymbol{\phi},\boldsymbol{\psi}} d\mu(\boldsymbol{\phi},\boldsymbol{\psi}) \Big( Q(\boldsymbol{\beta}|\boldsymbol{\phi}) + Q(\boldsymbol{\beta}|\boldsymbol{\psi}) \Big)$$

$$\leq 2\Delta_{\max},$$

where $Q(\cdot|\cdot) = \prod_p Q_p(\cdot|\cdot)$. Thus, the one-step total variation distance between the exact kernel and the proposed approximate kernel is upper bounded by

$$\|\mu\mathcal{T}(\boldsymbol{\beta}) - \mu\mathcal{T}_\eta(\boldsymbol{\beta})\|_{\text{TV}} = \frac{1}{2} \int_{\boldsymbol{\beta}} d\Omega(\boldsymbol{\beta}) \Big| \mu\mathcal{T}(\boldsymbol{\beta}) - \mu\mathcal{T}_\eta(\boldsymbol{\beta}) \Big| = \Delta_{\max}. \tag{57}$$

The following is a restatement of Lemma 3 in the supplementary file of Korattikara et al. (2014).

**Lemma 4.** *Consider two transition kernels $\mathcal{T}$ and $\mathcal{T}_\eta$, which yield stationary distributions $\pi$ and $\pi_\eta$, respectively. If $\mathcal{T}$ follows a contraction such that there is a constant $\rho \in [0,1)$ for all probability distributions $\mu$:*

$$\|\mu\mathcal{T} - \pi\|_{TV} \leq \rho\|\mu - \pi\|_{TV},$$

*and the uniform upper bound of the one step error between $\mathcal{T}$ and $\mathcal{T}_\eta$ follows that*

$$\|\mu\mathcal{T} - \mu\mathcal{T}_\eta\|_{TV} \leq \Delta, \forall \mu,$$

*where $\Delta \geq 0$ is a constant. Then the total variation distance between $\pi$ and $\pi_\eta$ is bounded by*

$$\|\pi - \pi_\eta\|_{TV} \leq \frac{\Delta}{1-\rho}.$$

Under the smoothness assumption C2 and the dissipative assumption C3, the geometric ergodicity of the continuous-time replica exchange Langevin diffusion to the invariant measure $\pi$ has been established (Chen et al., 2019). Combining the exponential convergence of the KL divergence (Deng et al., 2020) and the Pinsker's inequality (Csiszár & Körner, 2011), we have that for any probability density $\mu$, there exists a contraction parameter $\rho \in (0, 1)$ such that

$$\|\mu\mathcal{T} - \pi\|_{\text{TV}} \leq \rho\|\mu - \pi\|_{\text{TV}},$$

where $\rho$ depends on the spectral gap established in Raginsky et al. (2017) and the swap rate.

Applying Lemma 4, the total variation distance can be upper bounded by

$$\|\pi - \pi_\eta\|_{\text{TV}} \leq \frac{\Delta_{\max}}{1-\rho},$$

which concludes the proof of Theorem 2. □

### C.2.1 BIAS ANALYSIS FOR GENERALIZED WINDOWS IN BIG DATA PROBLEMS.

Following the empirical setup in section 5.1 (Deng et al., 2020), we run parallel tempering based on two gradient Langevin dynamics (GLD) with noisy energy estimators of different variances and generalized windows of different sizes. We fix the learning rate 0.003 and set the temperatures as 1 and 10, respectively. The swap condition follows from Eq.(16). We run the algorithm 3,000,000 iterations and also include a baseline by running the low-temperature process with 6,000,000 iterations.

We try different window sizes (2 and 5) and difference standard deviations (1, 2, and 3) for the noisy energy estimators. We observe in Fig.11 that simply running GLD (red curves) suffers from the local trap issues and consistently lead to the worst performance; the exact energy estimators, as shown in the cyan curves, yield crude estimations as well due to the inclusion of stopping time. By contrast, as the variance increases, we see a pattern that the bias tends to decrease and a good trade-off [†] is obtained at $sd = 3$, where the blue curves show remarkable approximations.

---

[†] Further increasing sd (or the variance) may yield fewer swaps and less accelerations.

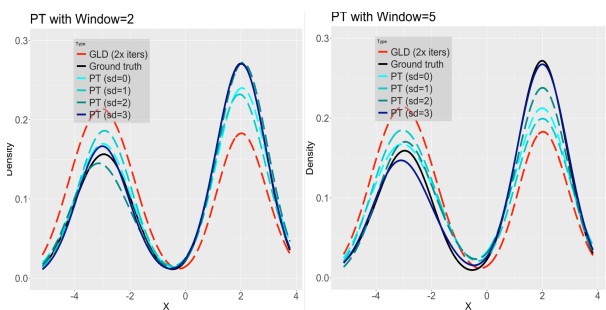

Figure 11: Bias analysis based on extended windows and nosiy energy estimators.

### C.3 WEAK ERROR

In this section, we show the weak error (Lemma 5 below) between the Replica exchange Langevin diffusion $(\boldsymbol{\beta}_t^{(1)}, \boldsymbol{\beta}_t^{(2)}, \cdots, \boldsymbol{\beta}_t^{(P)})$ and the Replica exchange stochastic gradient Langevin diffusion $(\widetilde{\boldsymbol{\beta}}_t^{(1)}, \widetilde{\boldsymbol{\beta}}_t^{(2)}, \cdots, \widetilde{\boldsymbol{\beta}}_t^{(P)})$. We denote $\mathbb{P} := \mathbb{P}^{\boldsymbol{W}} \times \mathbf{N}$, where $\mathbb{P}^{\boldsymbol{W}}$ is the infinite dimensional Wiener measure and $\mathbf{N}$ is the Poisson measure independent of $\mathbb{P}^{\boldsymbol{W}}$ and has some constant jump intensity. In our general framework below, the jump process $\alpha$ is introduced by swapping the diffusion matrix of the $P$ Langevin dynamics and the jump intensity is defined through the swap probability in the following sense, which ensures the independence of $\mathbb{P}^{\boldsymbol{W}}$ and $\mathbf{N}$ in each time interval $[i\eta, (i+1)\eta]$, for $i \in \mathbb{N}^+$. Without loss of generality, the replica exchange Langevin diffusion (reLD) is reformulated as below. For any fixed learning rate $\eta > 0$, we define

$$\begin{cases} d\boldsymbol{\beta}_t = -\nabla U(\boldsymbol{\beta}_t)dt + \Sigma(\alpha_t)d\boldsymbol{W}_t, \\ \mathbb{P}\left(\alpha(t) = j | \alpha(t - dt) = l, \boldsymbol{\beta}(\lfloor t/\eta \rfloor \eta) = \boldsymbol{\beta}\right) = aS(\boldsymbol{\beta})\eta \mathbf{1}_{\{t = \lfloor t/\eta \rfloor \eta\}} + o(dt), \text{ for } l \neq j, \end{cases} \quad (58)$$

where $\nabla U(\boldsymbol{\beta}) := \left(\nabla U(\boldsymbol{\beta}^{(1)}), \nabla U(\boldsymbol{\beta}^{(2)}), \cdots, \nabla U(\boldsymbol{\beta}^{(P)})\right)^{\mathsf{T}}$, and $\mathbf{1}_{t = \lfloor t/\eta \rfloor \eta}$ is the indicator function, i.e. for every $t = i\eta$ with $i \in \mathbb{N}^+$, given $\boldsymbol{\beta}(i\eta) = \boldsymbol{\beta}$, we have $\mathbb{P}\left(\alpha(t) = j | \alpha(t - dt) = l\right) = rS(\boldsymbol{\beta})\eta$, where $S(\boldsymbol{\beta})$ is defined as $\min\{1, S(\boldsymbol{\beta}^{(1)}, \boldsymbol{\beta}^{(2)})\}$ and $S(\boldsymbol{\beta}^{(1)}, \boldsymbol{\beta}^{(2)})$ is defined in (2). In this case, the Markov Chain $\alpha(t)$ is a constant on the time interval $[\lfloor t/\eta \rfloor \eta, \lfloor t/\eta \rfloor \eta + \eta)$ with some state in the finite-state space $\mathcal{M} = \{0, 1, \cdots, 2^{P-1} - 1\}$ and the generator matrix $Q$ follows,

$$\begin{cases} Q_{11} & = -aS(\boldsymbol{\beta}^{(1),(2)})\eta\delta(t - \lfloor t/\eta \rfloor \eta), \quad Q_{12} = aS(\boldsymbol{\beta}^{(1),(2)})\eta\delta(t - \lfloor t/\eta \rfloor \eta) \\ Q_{i(i-1)} & = aS(\boldsymbol{\beta}^{(i-1),(i)})\eta\delta(t - \lfloor t/\eta \rfloor \eta), \quad Q_{i(i+1)} = aS(\boldsymbol{\beta}^{(i),(i+1)})\eta\delta(t - \lfloor t/\eta \rfloor \eta) \\ Q_{ii} & = -aS(\boldsymbol{\beta}^{(i-1),(i)})\eta\delta(t - \lfloor t/\eta \rfloor \eta) - aS(\boldsymbol{\beta}^{(i),(i+1)})\eta\delta(t - \lfloor t/\eta \rfloor \eta) \\ Q_{P(P-1)} & = aS(\boldsymbol{\beta}^{(P-1),(P)})\eta\delta(t - \lfloor t/\eta \rfloor \eta), \quad Q_{PP} = -aS(\boldsymbol{\beta}^{(P-1),(P)})\eta\delta(t - \lfloor t/\eta \rfloor \eta), \end{cases}$$

where $1 < i < P$, and all the other terms in $Q$ remain zero. We denote $\delta(\cdot)$ as a Dirac delta function. Furthermore, for general adjacent swap with odd iterations (denoted as $Q^O$), and even iterations (denoted as $Q^E$), the generator matrices $Q^O$ and $Q^E \in \mathbb{R}^{(Pd) \times (Pd)}$ have the following form,

$$Q^O = \begin{pmatrix} Q_{11} & Q_{12} & 0 & 0 & \cdots \\ Q_{21} & -Q_{21} & 0 & 0 & 0 \\ 0 & 0 & -Q_{34} & Q_{34} & 0 \\ 0 & 0 & Q_{43} & -Q_{43} & 0 \\ 0 & 0 & 0 & 0 & \cdots \\ & & \cdots & & \cdots \end{pmatrix}, \quad Q^E = \begin{pmatrix} 0 & 0 & 0 & 0 & 0 \\ 0 & -Q_{23} & Q_{23} & 0 & 0 & \cdots \\ 0 & Q_{32} & -Q_{32} & 0 & 0 & 0 \\ 0 & 0 & 0 & -Q_{45} & Q_{45} & \cdots \\ 0 & 0 & 0 & Q_{54} & -Q_{54} & 0 \\ & & \cdots & & & \cdots \end{pmatrix},$$

For the odd iteration case, the diffusion matrix for the first two chains $\boldsymbol{\beta}^{(1)}$ and $\boldsymbol{\beta}^{(2)}$ has the following form according to the state of $\Sigma(\alpha_t)$, i.e. $(\Sigma(0), \Sigma(1)) :=$

$$\left\{ \begin{pmatrix} \sqrt{2\tau^{(1)}}\mathbf{I}_d & 0 & \cdots \\ 0 & \sqrt{2\tau^{(2)}}\mathbf{I}_d & \cdots \\ \cdots & \cdots & \cdots \end{pmatrix}_{(Pd) \times (Pd)}, \begin{pmatrix} \sqrt{2\tau^{(2)}}\mathbf{I}_d & 0 & \cdots \\ 0 & \sqrt{2\tau^{(1)}}\mathbf{I}_d & \cdots \\ \cdots & \cdots & \cdots \end{pmatrix}_{(Pd) \times (Pd)} \right\}, \quad \text{where}$$

the rest of the diffusion matrix remain unchanged. Based on the observation of the transition matrix of $Q^O$, and $Q^E$, there is no interaction between every pairs (each pair has two chains) in

each iteration, we will prove the weak error for only two chains. We can then add different square blocks together as shown in $Q^O$ and $Q^E$. From now on, we restrict to the two chain case with $\boldsymbol{\beta} = (\boldsymbol{\beta}^{(1)}, \boldsymbol{\beta}^{(2)})$. Next, we denote $\widetilde{\boldsymbol{\beta}}$ as the Replica exchange stochastic gradient Langevin diffusion, for the same learning rate $\eta > 0$ as above, we have

$$
\begin{cases}
d\widetilde{\boldsymbol{\beta}}_t^\eta = -\nabla\widetilde{U}(\widetilde{\boldsymbol{\beta}}_{\lfloor t/\eta\rfloor\eta}^\eta)dt + \Sigma(\widetilde{\alpha}_{\lfloor t/\eta\rfloor\eta})d\boldsymbol{W}_t, \\[2mm]
\mathbb{P}\left(\widetilde{\alpha}(t) = j | \widetilde{\alpha}(t - dt) = l, \widetilde{\boldsymbol{\beta}}(\lfloor t/\eta\rfloor\eta) = \widetilde{\boldsymbol{\beta}}\right) = \widetilde{S}(\widetilde{\boldsymbol{\beta}})\eta\mathbf{1}_{\{t=\lfloor t/\eta\rfloor\eta\}} + o(dt), \ \text{for} \ l \neq j,
\end{cases}
\tag{59}
$$

where $\nabla\widetilde{U}(\boldsymbol{\beta}) := \begin{pmatrix} \nabla\widetilde{U}(\boldsymbol{\beta}^{(1)}) \\ \nabla\widetilde{U}(\boldsymbol{\beta}^{(2)}) \end{pmatrix}$ and $\widetilde{S}(\widetilde{\boldsymbol{\beta}}) = \min\{1, \widetilde{S}_{\eta,m,n}(\widetilde{\boldsymbol{\beta}}^{(1)}, \widetilde{\boldsymbol{\beta}}^{(2)})\}$ and $\widetilde{S}_{\eta,m,n}(\widetilde{\boldsymbol{\beta}}^{(1)}, \widetilde{\boldsymbol{\beta}}^{(2)})$ is

shown in Eq.(16). The distribution of process $(\widetilde{\boldsymbol{\beta}}_t)_{0\leq t\leq T}$ is denoted as $\mu_T := \mathbb{P}^{\widetilde{G}} \times \mathbf{N}^{\widetilde{S}}$, where $\widetilde{\alpha}$ is a Poisson process with jump intensity $a\widetilde{S}(\widetilde{\boldsymbol{\beta}})\eta\delta(t - \lfloor t/\eta\rfloor\eta)$ on the time interval $[\lfloor t/\eta\rfloor\eta, \lfloor t/\eta\rfloor\eta + \eta)$.

For any open set $\mathcal{U} \subset \mathbb{R}^{2d}$, which is contained in a large enough open ball with radius $R$ centered at $\boldsymbol{\beta}(0)$, we define stopping time

$$
\tau_{\mathcal{U}} = \inf\{t \geq 0 | (\boldsymbol{\beta}(t), \alpha(t)) \notin \mathcal{U} \times \mathcal{M}\}.
\tag{60}
$$

**Lemma 5.** *Assume the smoothness assumption C2 holds. For any large enough open set $\mathcal{U}$, we have*

$$
|\mathbb{E}^{x,i}[h(\boldsymbol{\beta}(T \wedge \tau_{\mathcal{U}}), \alpha(T \wedge \tau_{\mathcal{U}}))] - \mathbb{E}^{x,i}[h(\tilde{\boldsymbol{\beta}}(T \wedge \tau_{\mathcal{U}}), \widetilde{\alpha}(T \wedge \tau_{\mathcal{U}}))]| = \mathcal{O}(\eta).
$$

*for any continuous differentiable and polynomial growth function $h$.*

*Proof.* We denote $\boldsymbol{\beta}_t = (\boldsymbol{\beta}_t^{(1)}, \boldsymbol{\beta}_t^{(2)})$, which follows the dynamics below,

$$
d\boldsymbol{\beta}_t = -\nabla U(\boldsymbol{\beta}_t)dt + \Sigma(\alpha_t)dW_t
$$

Denote

$$
u(\boldsymbol{\beta}_t = x, t, i) = \mathbb{E}^{x,i}[h(\boldsymbol{\beta}_{T\wedge\tau_{\mathcal{U}}}, \alpha_{T\wedge\tau_{\mathcal{U}}})|(\boldsymbol{\beta}_t, \alpha_t) = (x, i)], \quad i \in \mathcal{M}.
$$

For $t < T \wedge \tau_{\mathcal{U}}$, applying Feynman-Kac formula (see e.g. Baran et al. (2013)), we have

$$
\begin{cases}
\partial_t u(x,t,i) + \mathcal{L}u(x,t,i) = 0, \\
\mathcal{L}u(x,t,i) = tr(\Sigma\Sigma^T\nabla^2 u(x,t,i)) - \langle\nabla u(x,t,i), \nabla U\rangle + \mathcal{Q}(x,t)u(x,t,\cdot)(i)
\end{cases}
\tag{61}
$$

Applying generalized Itô formula, for each $i \in \mathcal{M}$ we have

$$
h(\boldsymbol{\beta}_t, \alpha_t) - h(\boldsymbol{\beta}_0, \alpha_0) = \int_0^t \mathcal{L}g(\boldsymbol{\beta}_s, \alpha_s)ds + M_1(t) + M_2(t),
\tag{62}
$$

where $M_1 + M_2$ is a local martingale. See details in (Yin & Zhu, 2010; Skorokhod, 2009). We thus get, for $k\eta \leq t < (k+1)\eta < T \wedge \tau_{\mathcal{U}}$,

$$
\mathbb{E}^{x,i}[h(\boldsymbol{\beta}_t, \alpha_t)] - h(x, i) = \mathbb{E}^{x,i}\int_0^t \mathcal{L}h(\boldsymbol{\beta}_s, \alpha_s)ds.
$$

For $0 < t_1 < t_2 < \cdots, t_N = T \wedge \tau_{\mathcal{U}}$, with $t_k = k\eta$, we next derive the Itô formula for $h(\widetilde{\boldsymbol{\beta}}_t, \alpha_t)$.

Note that $\boldsymbol{\beta}$ and $\widetilde{\boldsymbol{\beta}}$ are defined by using the same $\mathbb{P}$-Brownian motion $\boldsymbol{W}$, but with two different jump intensity on the time interval $[\lfloor t/\eta\rfloor\eta, \lfloor t/\eta\rfloor\eta + \eta)$. We have the following approximation,

$$
\widetilde{\boldsymbol{\beta}}_{t_k} - \widetilde{\boldsymbol{\beta}}_{t_{k-1}} = -\nabla\widetilde{U}(\widetilde{\boldsymbol{\beta}}_{t_{k-1}})\eta + \Sigma(\widetilde{\alpha}_{t_{k-1}})\mathcal{N}(0, \eta).
$$

For stochastic noise $\xi_{t_{k-1}}$, with $\mathbb{E}[\xi_{t_{k-1}}] = 0$, we rewrite the above approximation as below,

$$
\widetilde{\boldsymbol{\beta}}_{t_k} - \widetilde{\boldsymbol{\beta}}_{t_{k-1}} = -(\nabla U(\widetilde{\boldsymbol{\beta}}_{t_{k-1}}) + \xi_{t_{k-1}})\eta + \Sigma(\widetilde{\alpha}_{t_{k-1}})\mathcal{N}(0, \eta).
$$

For notation convenience, for $t_{k-1} \leq t < t_k$, we keep the following convention,

$$
\widetilde{\Sigma}(\widetilde{\boldsymbol{\beta}}_t) := \Sigma(\widetilde{\boldsymbol{\beta}}_{\lfloor t/\eta\rfloor\eta}^\eta) = \Sigma(\widetilde{\boldsymbol{\beta}}_{t_{k-1}}^\eta), \quad -\nabla\widetilde{U}(\widetilde{\boldsymbol{\beta}}_t) := -\nabla\widetilde{U}(\Sigma(\widetilde{\boldsymbol{\beta}}_{\lfloor t/\eta\rfloor\eta}^\eta)) = -\nabla\widetilde{U}(\widetilde{\boldsymbol{\beta}}_{t_{k-1}}).
$$

For any $t_{k-1} \leq t < t_k$, we apply Itô formula for $u(\widetilde{\boldsymbol{\beta}}_t, t, i)$,

$$du(\widetilde{\boldsymbol{\beta}}_t, t, i) = (\partial_t u + \widetilde{\mathcal{L}} u(\widetilde{\boldsymbol{\beta}}_t, i)dt + \Sigma(\widetilde{\alpha}_{\lfloor t/\eta \rfloor \eta})\frac{\partial u}{\partial x}(\widetilde{\boldsymbol{\beta}}_t)dW_t + d\widetilde{M}_1(t) + d\widetilde{M}_2(t), \qquad (63)$$

$$= [(\widetilde{\mathcal{L}} - \mathcal{L})u(\widetilde{\boldsymbol{\beta}}_t, t, i)]dt + \Sigma(\widetilde{\alpha}_{\lfloor t/\eta \rfloor \eta})\frac{\partial u}{\partial x}(\widetilde{\boldsymbol{\beta}}_t)dW_t + d\widetilde{M}_1(t) + d\widetilde{M}_2(t),$$

where the last equality follows from (61), and for $t_{k-1} \leq t < t_k$,

$$\widetilde{\mathcal{L}} u(\widetilde{\boldsymbol{\beta}}_t, t, i) = tr(\widetilde{\Sigma}\widetilde{\Sigma}^T \nabla^2 u(\widetilde{\boldsymbol{\beta}}_t, t, i)) - \langle \nabla \widetilde{U}(\widetilde{\boldsymbol{\beta}}_t), \nabla u \rangle + \widetilde{\mathcal{Q}}(\widetilde{\boldsymbol{\beta}}_t, t)u(\widetilde{\boldsymbol{\beta}}_t, t, \cdot)(i).$$

Evaluate the integral on time interval $[0, T \wedge \tau_{\mathcal{U}}]$, and take expectation $\mathbb{E}^{x,i}$, we have

$$\mathbb{E}^{x,i}[u(\widetilde{\boldsymbol{\beta}}_{T \wedge \tau_{\mathcal{U}}}, T \wedge \tau_{\mathcal{U}}, i) - u(\boldsymbol{\beta}_0, 0, i)] = \int_0^{T \wedge \tau_{\mathcal{U}}} \mathbb{E}^{x,i}[(\widetilde{\mathcal{L}} - \mathcal{L})u(\widetilde{\boldsymbol{\beta}}_t, t)]dt$$

$$= \mathcal{I}_1 + \mathcal{I}_2 + \mathcal{I}_3.$$

Where we have

$$\mathcal{I}_1 := -\int_0^{T \wedge \tau_{\mathcal{U}}} \mathbb{E}^{x,i}[\langle \nabla \widetilde{U}(\widetilde{\boldsymbol{\beta}}_t), \nabla u(\widetilde{\boldsymbol{\beta}}_t, t) \rangle - \langle \nabla U(\widetilde{\boldsymbol{\beta}}_t), \nabla u(\widetilde{\boldsymbol{\beta}}_t, t) \rangle]dt,$$

$$\mathcal{I}_2 := \int_0^{T \wedge \tau_{\mathcal{U}}} \mathbb{E}^{x,i}[tr(\widetilde{\Sigma}\widetilde{\Sigma}^T \nabla^2 u(\widetilde{\boldsymbol{\beta}}_t, t)) - tr(\Sigma \Sigma^T \nabla^2 u(\widetilde{\boldsymbol{\beta}}_t, t))]dt,$$

$$\mathcal{I}_3 := \int_0^{T \wedge \tau_{\mathcal{U}}} \mathbb{E}^{x,i}[\widetilde{\mathcal{Q}}(\widetilde{\boldsymbol{\beta}}_t, t)u(\widetilde{\boldsymbol{\beta}}_t, t, \cdot) - \mathcal{Q}(\widetilde{\boldsymbol{\beta}}_t, t)u(\widetilde{\boldsymbol{\beta}}_t, t, \cdot)]dt.$$

In the next step, we introduce a sequence of stopping time based on our definition of process $\boldsymbol{\beta}$ and $\widetilde{\boldsymbol{\beta}}$. For $j \in \mathbb{N}^+$, we denote $\zeta_j's$ as a stopping times defined by $\zeta_{j+1} := \inf\{t > \zeta_j : \alpha(t) \neq \alpha(\zeta_j)\}$ and $N(T) = \max\{n \in \mathbb{N} : \zeta_n \leq T\}$. According to the definition of process $\boldsymbol{\beta}$, we have $\zeta_j = l\eta$, for some $l \in \mathbb{N}^+$. Similarly, we have stopping time $\widetilde{\zeta}_{j+1} := \inf\{t > \widetilde{\zeta}_j : \widetilde{\alpha}(t) \neq \widetilde{\alpha}(\zeta_j)\}$ associated with process $\widetilde{\boldsymbol{\beta}}$, and $\widetilde{\alpha}(t)$ follows the same trajectory of $\alpha(t)$. We thus have the following estimates for the terms $\mathcal{I}_1, \mathcal{I}_2,$ and $\mathcal{I}_3$.

**Estimate of $\mathcal{I}_1$:** as for $\mathcal{I}_1$, $-\nabla U(\cdot)$ does not depend on $\alpha$, we get

$$\mathcal{I}_1 = -\sum_{k=0}^{K-1} \int_{t_k}^{t_{k+1}} \mathbb{E}^{x,i}[\langle (\nabla U(\widetilde{\boldsymbol{\beta}}_{t_k}) + \xi_{t_k} - \nabla U(\widetilde{\boldsymbol{\beta}}_t)), \nabla u(\widetilde{\boldsymbol{\beta}}_t, t) \rangle]dt$$

$$= -\sum_{k=0}^{K-1} \int_{t_k}^{t_{k+1}} \mathbb{E}^{x,i}[\langle (\nabla U(\widetilde{\boldsymbol{\beta}}_{t_k}) - \nabla U(\widetilde{\boldsymbol{\beta}}_t)), \nabla u(\widetilde{\boldsymbol{\beta}}_t, t) \rangle]dt,$$

where we use the assumption $\mathbb{E}^{x,i}[\xi_{t_k}] = 0$, and independent property of the noise $\xi_{t_k}$. For $t_k \leq t < t_{k+1}$, let

$$\rho_1(\widetilde{\boldsymbol{\beta}}_t, t) = \langle (\nabla U(\widetilde{\boldsymbol{\beta}}_{t_k}) - \nabla U(\widetilde{\boldsymbol{\beta}}_t)), \nabla u(\widetilde{\boldsymbol{\beta}}_t, t) \rangle,$$

applying Itô formula (same as (63) above), taking expectation, and using the fact $\widetilde{\mathcal{Q}}\rho_1(\widetilde{\boldsymbol{\beta}}_t, t) = a\widetilde{S}(\widetilde{\boldsymbol{\beta}}_t^{(1)}, \widetilde{\boldsymbol{\beta}}_t^{(2)})[\rho_1((\boldsymbol{\beta}_t^{(1)}, \boldsymbol{\beta}_t^{(2)}), t) - \rho_1((\boldsymbol{\beta}_t^{(2)}, \boldsymbol{\beta}_t^{(1)}), t)] = 0$, we get

$$\frac{d\mathbb{E}^{x,i}[\rho_1(\widetilde{\boldsymbol{\beta}}_t, t)]}{dt} = \mathbb{E}^{x,i}[\partial_t \rho_1(\widetilde{\boldsymbol{\beta}}_t, t) + tr(\widetilde{\Sigma}\widetilde{\Sigma}^T \nabla^2 \rho_1(\widetilde{\boldsymbol{\beta}}_t, t)) - \langle \nabla \widetilde{U}(\widetilde{\boldsymbol{\beta}}_t), \nabla \rho_1(\widetilde{\boldsymbol{\beta}}_t, t) \rangle] \leq C_k,$$

where the bound follows from the Weierstras theorem, and the same idea is also adopted in Sato & Nakagawa (2014) for diffusion process without jumps. We end up with,

$$\mathbb{E}^{x,i}[\rho_1(\widetilde{\boldsymbol{\beta}}_t, t)] \leq C_k \eta,$$

which implies

$$\mathcal{I}_1 \leq T\eta C_{max}.$$

**Estimate of** $\mathcal{I}_2$: for each time interval $t_k \leq t < t_{k+1}$, falling between two stopping time, the matrix $\Sigma(\alpha(t)) = \widetilde{\Sigma}(\widetilde{\alpha}(t))$ remains constant, and $tr(\Sigma\Sigma^T \nabla^2 u) = tr(\widetilde{\Sigma}\widetilde{\Sigma}^T \nabla^2 u) = \sum_{j=0}^{P} \tau_j \Delta_{\widetilde{\beta}_j} u$, we have

$$\mathcal{I}_2 = 0.$$

**Estimate of** $\mathcal{I}_3$: we rewrite the time interval $[0, T \wedge \tau_{\mathcal{U}}]$ as the summation of the stopping times, i.e. $\int_0^{T \wedge \tau_{\mathcal{U}}} = \sum_{l=0}^{N(T \wedge \tau_{\mathcal{U}})-1} \int_{\zeta_l}^{\zeta_{l+1}}$, we thus have

$$
\begin{aligned}
\mathcal{I}_3 &:= \int_0^{T \wedge \tau_{\mathcal{U}}} \mathbb{E}^{x,i}[\widetilde{\mathcal{Q}}(\widetilde{\boldsymbol{\beta}}_t, t)u(\widetilde{\boldsymbol{\beta}}_t, t, \cdot) - \mathcal{Q}(\widetilde{\boldsymbol{\beta}}_t, t)u(\widetilde{\boldsymbol{\beta}}_t, t, \cdot)]dt \\
&= \sum_{l=0}^{N(T \wedge \tau_{\mathcal{U}})-1} \int_{\zeta_l}^{\zeta_{l+1}} \mathbb{E}^{x,i}[\widetilde{\mathcal{Q}}(\widetilde{\boldsymbol{\beta}}_t, t)u(\widetilde{\boldsymbol{\beta}}_t, t, \cdot) - \mathcal{Q}(\widetilde{\boldsymbol{\beta}}_t, t)u(\widetilde{\boldsymbol{\beta}}_t, t, \cdot)]dt.
\end{aligned}
$$

Recall that,
$$\mathcal{Q}(x)u(x, \cdot)(i) = \sum_{j \in \mathcal{M}, j \neq i} Q_{ij}(u(x, j) - u(x, i)),$$

and we denote $\widetilde{Q}_{ij}$ as the components of the generator matrix $\widetilde{Q}$ of $\widetilde{\boldsymbol{\beta}}$. For $\zeta_l \leq t < \zeta_{l+1}$, we denote

$$
\begin{aligned}
\rho_3^l(\widetilde{\boldsymbol{\beta}}_t, t) &:= \widetilde{\mathcal{Q}}(\widetilde{\boldsymbol{\beta}}_t, t)u(\widetilde{\boldsymbol{\beta}}_t, t, \cdot)(\widetilde{\alpha}(\tau_l)) - \mathcal{Q}(\widetilde{\boldsymbol{\beta}}_t, t)u(\widetilde{\boldsymbol{\beta}}_t, t, \cdot)(\alpha(\tau_l)) \\
&= \sum_{j \in \mathcal{M}, j \neq \alpha(\tau_l)} (\widetilde{Q}_{\alpha(\tau_l),j}(\widetilde{\boldsymbol{\beta}}_t) - Q_{\alpha(\tau_l),j}(\widetilde{\boldsymbol{\beta}}_t))(u(\widetilde{\boldsymbol{\beta}}_t, t)(\alpha(j) - u(\widetilde{\boldsymbol{\beta}}_t, t)\alpha(\tau_l))).
\end{aligned}
$$

In particular, for the two chain case, $\mathcal{M} = \{0, 1\}$. For any $t \leq T \wedge \tau_{\mathcal{U}}$, there exists large enough open ball with radius $R$, denoted as $\mathcal{U} \subset \mathcal{B}_{(\boldsymbol{\beta}(0),R)}$, such that $\mathcal{U} \subset \mathcal{B}_{(\boldsymbol{\beta}(0),R)}$, which implies the uniform bound of $u(\boldsymbol{\beta}_t, t) \leq C$, for $t \leq T \wedge \tau_{\mathcal{U}}$. We thus get the estimates

$$
\begin{aligned}
|\mathcal{I}_3| &\leq \sum_{l=0}^{N(T \wedge \tau_{\mathcal{U}})-1} \int_{\zeta_l}^{\zeta_{l+1}} \mathbb{E}^{x,i}[|\rho_3^l(\widetilde{\boldsymbol{\beta}}_t, t)|] \\
&\leq 2C \sum_{l=0}^{N(T \wedge \tau_{\mathcal{U}})-1} \int_{\zeta_l}^{\zeta_{l+1}} \sum_{j \in \mathcal{M}, j \neq \alpha(\tau_l)} \mathbb{E}^{x,i}[|(\widetilde{Q}_{\alpha(\tau_l),j}(\widetilde{\boldsymbol{\beta}}_t) - Q_{\alpha(\tau_l),j}(\widetilde{\boldsymbol{\beta}}_t))|] \\
&\leq 2Ca\eta \sum_{l=0}^{N(T \wedge \tau_{\mathcal{U}})-1} \int_{\zeta_l}^{\zeta_{l+1}} \sum_{j \in \mathcal{M}, j \neq \alpha(\tau_l)} \mathbb{E}^{x,i}[|(\widetilde{S}_{\alpha(\tau_l),j}(\widetilde{\boldsymbol{\beta}}_t) - S_{\alpha(\tau_l),j}(\widetilde{\boldsymbol{\beta}}_t))|] \\
&\leq 2CKa\eta.
\end{aligned}
$$

where the last inequality follows from the fact that $S, \widetilde{S} \leq 1$, and $N(T \wedge \tau_{\mathcal{U}}) \leq K$. Combing all the estimates in $\mathcal{I}_1, \mathcal{I}_2, \mathcal{I}_3$, the proof is thus completed.

$\square$

