# OpenReview forum: "Non-reversible Parallel Tempering for Uncertainty Approximation in Deep Learning"
_ICLR.cc/2022/Conference — ICLR 2022 Submitted_

### Official Review · Reviewer_kvcG · 2021-10-25

**Correctness:** 3
**Technical Novelty And Significance:** 3
**Empirical Novelty And Significance:** 3
**Recommendation:** 5
**Confidence:** 3

**Main Review:**

The idea considered is very interesting and it demonstrates a lot of promises. The theoretical components seem to be strong too. I’ll be happy to be corrected and increase my rating, but I’d love to see more integration of various theoretical components, as well as how their implications get reflected in the empirical results. For example:
* Is there any theoretical guide on the choice of $a$? By the way, what is the relation between $a$ and $\mathbb{S}$?
* If I understood correctly, Theorem 2 is just for one chain, not all the $P$ chains. When there are $P>1$ chains, don't errors add up? How does that scale with $P$?
* How can Theorem 1 and Theorem 2 be combined to produce a theoretical guarantee of the final algorithm? It will be great if notions built around `round trip’ get translated into final iteration complexity.
* Can wall clock counts be added to the experiments? Since improved efficiency seems to be a key contribution, it is a little odd to only show improved accuracy.
* Can results for various W values be compared in the experiments, so that the theoretically optimal value could be justified?
* Reading `We choose ResNet20, ResNet32, and ResNet56 (He et al., 2016) and train the models on CIFAR100.’ can I assume Bayesian neural network is used based on these architectures? What is the prior?


**Summary Of The Paper:**

This paper considers sampling multi-modal distributions by the powerful parallel tempering methodology. An existing exchange scheme, DEO (deterministic even-odd), is improved by generalizing even and odd chain indices to even and odd windows of width W. Some nontrivial theoretical and empirical results are provided.

**Summary Of The Review:**

This is a very interesting paper. Since the treatment is rather nontrivial and I feel it deserves appreciation, I’d like to see just a little more about how different theoretical and empirical ingredients of this work mesh together.

---

> ### Author Response · Authors · 2021-11-17
> **Response to Reviewer kvcG**
>
> We appreciate the valuable comments.
>
> $\newline$
>
> Q1. Theoretical guidance of $a$ and the relation between a and $\mathbb{S}$.
>
> As shown right after Eq.(16) in page 13 of our work (or see Choice of swapping intensity in page 16 of [1]), the swap intensity $a$ should be equal to or smaller than $\frac{1}{\min\\{\eta_p, \eta_{p+1}\\}}$.
>
> Note that $\mathbb{S}$ is not much related to $a$. For example, the simulation of Gaussian mixture example in section 5.1 of [2] shows that setting $\mathbb{S}=0.4\\%$  leads to the best approximations based on a limited computational budget. By contrast, following the exact bias-corrected swap rate ends up with almost no swaps and fails in the accelerations; setting $\mathbb{S}$ too high is not acceptable either due to the large bias.
>
> We suggest tuning $\mathbb{S}$ and balancing between acceleration and accuracy directly through cross-validation, which is standard in deep learning.
>
>
>
> [1] Accelerating Nonconvex Learning via Replica Exchange Langevin Diffusion. ICLR'19. arxiv.org/pdf/2007.01990.pdf
>
>
> [2] Non-convex Learning via Replica Exchange Stochastic Gradient MCMC. ICML'20.
>
>
> $\newline$
>
> Q2. Is theorem 2 based on multiple chains or not?
>
> Yes. We acknowledge that the current approximation error may become worse as more chains are proposed. The study of whether the error $\Delta_{\max}$ is bounded or not as $P\rightarrow \infty$ is interesting and will be left as future works.
>
> $\newline$
>
> Q3. Can we combine Theorem 1 and 2 and get some final iteration complexity.
>
> Unfortunately, we cannot. The first challenge is that although the continuous-time Markov jump operator yields a Dirichlet term with an acceleration term, it is quite hard to identify the lower bound of it in general. The acceleration based on the spectral gap for a specific Gaussian mixture example with equal weights has been quantified in [1] but the general theory is still missing. Devising a general theory on the accelerations goes beyond this work.
>
>
> \begin{equation}
> \begin{split}
> {\cal E_{S}} (f)={\cal E} (f) +\underbrace{\frac{a}{2}\int S(\beta^{(1)},\beta^{(2)})\cdot (f(\beta^{(2)},\beta^{(1)})-f(\beta^{(1)},\beta^{(2)}))^2d\pi(\beta^{(1)},\beta^{(2)})}_{\text{acceleration term}}
> \end{split}
> \end{equation}
>
> The second challenge is that the algorithm is mainly proposed for a user-friendly purpose in deep learning tasks with low-accuracy demands and the upper bound of the bias can be arbitrarily bad in the worst case, although it works well in practice.
>
>
> Due to these two complications, we cannot easily produce complexity results as in [2]
>
>
>
> [1] Jing Dong and Xin T. Tong.   Spectral Gap of Replica Exchange Langevin Diffusion on MixtureDistributions. ArXiv 2006.16193v2, July 2020.
>
> [2] Global Convergence of Langevin Dynamics Based Algorithms for Nonconvex Optimization. NIPS'18.
>
> $\newline$
>
> Q4. Discussions of the wall clock time.
>
> In shared-memory settings, our algorithm is as fast as the mSGD algorithm with parallel runs.
>
> In big data problems based on distributed-memory scenarios, a pretty small target swap rate $\mathbb{S}$ is also required due to the large correction terms and the large optimal window size, such as $W=626$ in CIFAR100 in section 5.2, is inevitable, which has significantly reduced the communication cost. It is hence still as efficient as the standard mSGD algorithm.
>
> $\newline$
>
> Q5. Study of different W.
>
> We have provided the corresponding result, which shows in Fig 4(a) and Fig 7(a) that the empirical optimal window size matches the theoretical optimal.
>
> $\newline$
>
> Q6. Architectures and priors for training Bayesian neural networks
>
> Our Bayesian neural networks are based on ResNet model architectures, which are quite standard in CIFAR100 datasets. Other model structures can be also adopted.
>
> The prior is Gaussian, which is implemented based on the weight decay of scale 5e-4 in the average of data log-likelihood settings.

---

### Official Review · Reviewer_mbau · 2021-11-02

**Correctness:** 2
**Technical Novelty And Significance:** 4
**Empirical Novelty And Significance:** 4
**Recommendation:** 3
**Confidence:** 4

**Main Review:**

1. The generalized DEO scheme is interesting and the O(P logP) expected round trip time for finite number of chains and non-diminishing rejection rate is novel. However, DEO* doesn't converge to the target distribution. The following example constitutes a counterexample of soundness of DEO*.

Consider an energy function $f(x) = 8 x^2$ on the domain $[-1,1]$. We construct only two chains with temperatures $T_1=1$, $T_2=\infty$. Two posterior should be $p_1$ as a truncated Gaussian distribution highly concentrated around origin, $p_2$ be a uniform distribution. Let local exploration kernel to be identity (e.g. SGLD with $0$ step size). Let window size to be large enough, such that swap will happens in one window with high probability. Initialize two chains with target distributions $p_1$ and $p_2$, then run DEO* with one even window and one odd window. The output distribution is no longer $p_1$ and $p_2$.

2. This paper is clear until section 4 where DEO*-SGD method is proposed. Although practical, DEO*-SGD contains multiple approximations that require careful investigation:
+ a. While using SGD as approximated local exploration kernel in (6), it seems only gradient noise is considered. However, discretization error is another source of noise. More specifically, for Langevin dynamics $d\beta_t = -\nabla U(\beta_t) + \sqrt{2} dW_t$, the discretization error for Euler discretization is $\int_0^\eta \nabla U(\beta_t) dt-\eta\nabla U(\beta_k)$. How does this discretization error affect the distribution and temperature?
+ b. on second line at page 5, the authors state that "$\beta_k$ converges approximately to an invariant distribution, where the underlying temperature linearly depends on the learning rate $\eta$". What is this invariant distribution? Does different step size induce different invariant distribution?
+ c. With update rule in (12), can we ensure that $\eta^{(p)}$ are always ordered?

3. The experiments also need to be amended.
+ a. Important baseline (E.g. Deep Ensemble) is missing.
+ b. No uncertainty approximation quality (e.g. expected calibration error) is reported.
+ c. Figure 7.c has incorrect y-axis value.
+ d. experiment code for CIFAR100 experiment is not provided.

**Summary Of The Paper:**

This paper proposes generalized DEO scheme (DEO*) that extends DEO with a window where at most one exchange could happen within one window. This paper also extend existing analysis on expected round trip time into DEO* and authors show that O(P logP) expected round trip time could be achieved even for finite number of chains and non-diminishing rejection rate. This paper also proposes multiple approximation to make DEO* practical: The exploration chains substitute gaussian noise with gradient noise, where noise magnitude is controlled with step size. Deterministic swap condition is used to avoid specific temperature. Adaptive learning rates and adaptive correction buffers are adopted to tune acceptance rate. The authors evaluate DEO* on ResNet model over CIFAR100.

**Summary Of The Review:**

The proposed method does reduce expected round trip time but fails to converge. The practical method is not well understood and experiments need to be amended.

---

> ### Author Response · Authors · 2021-11-17
> **Response**
>
>
> We appreciate your valuable questions.
>
> **Q1. DEO$_{\star}$ doesn't converge to the invariant distribution.**
>
> We kindly disagree with your argument. Before the approximation analysis in section 4, the DEO$_{\star}$ scheme based on Langevin diffusion with the Metropolis acceptance-rejection swap rule is just like running the DEO scheme first and then freezing the rest of the swaps in the window.  The **reversibility based on Metropolis acceptance-rejection rules is always satisfied**, no matter whether there are multiple swaps, one swap, or no swap.
>
>
> Setting the ill-behaved temperature $T_2=\infty$ requires the learning rate $\eta_2=0$ to avoid infinite numerical errors (see Proposition 3.1 on page 9 of [1]). **We don't know if it is a well-defined process in the counterexample**. Nevertheless, the Metropolis acceptance-rejection rule still maintains reversibility.
>
> $\textcolor{red}{\text{Authors' remark on Nov. 22, after discussions with the reviewer, we acknowledge that the distribution may be affected due to the inclusion of stopping time. }}$
>
> $\textcolor{red}{\text{Fortunately, the bias can be controlled and becomes smaller in big data problems. The details will be presented in another thread.}}$
>
> [1] Non-Convex Learning via Stochastic Gradient Langevin Dynamics: A Nonasymptotic Analysis. COLT'17. arxiv.org/pdf/1702.03849.pdf
>
> $\newline$
>
> Q2. How does the discretization affect the distribution?
>
> A larger learning rate (and a larger temperature) leads to a larger discretization error. Please see our response in Lemma 5 on page 25. We will clarify the relation between $\Delta_{\max}$ and $(\eta, \mathbb{C})$ in the next revision.
>
> $\newline$
>
> Q3. Does a different step size of SGD induce a different invariant distribution?
>
> Yes. By assuming assumption C4 on page 22 (or assumption 2 in [1]), SGD with different step sizes is equivalent to gradient Langevin dynamics with different temperatures and step sizes. Similar results are shown in Theorem 1 of [1] such that an optimal step size exists to minimize the KL divergence between the stationary distribution proposed by SGD and the target posterior. In particular, a larger step size is more appropriate for exploration and a smaller one is suitable for exploitation.
>
>
> [1] Stochastic Gradient Descent as Approximate Bayesian Inference. JMLR. 2017.
>
> $\newline$
>
> Q4. Can we ensure that $\eta^{(p)}$ is always ordered?
>
> As long as the stability of the stochastic approximation process is guaranteed, we can ensure the sequence of step sizes is monotone increasing with probability 1. Admittedly, theoretical analysis on the upper bound of step size to maintain stability goes beyond this paper and will be left as future works. Similar ideas have been achieved in section 2.2 of [1].
>
> Empirically, given sufficiently small step sizes for stochastic approximation, the monotone order of learning rates is always guaranteed, as shown in the experiments.
>
> [1] Towards Optimal Scaling of Metropolis coupled Markov Chain Monte Carlo. Statistics and Computing, 2011.

---

> > ### Author Response · Authors · 2021-11-17
> > **Concerns on the experiments**
> >
> > Q5.1. The important baseline of the deep ensemble is missing.
> >
> > We do have provided those baselines. Please check SGLD$\times$P16 in Figure 5 (b) and mSGD$\times$P10 in Table 2.
> >
> >
> > $\newline$
> >
> > Q5.b. Criterion based on the expected calibration error (ECE) is not provided.
> >
> > As shown in [1], there are lots of criteria to evaluate uncertainty approximation, such as ECE, negative log likelihood, and entropy of out-of-distribution samples. We will include more criteria for uncertainty approximation in the next revision.
> >
> > [1]. A Simple Baseline for Bayesian Uncertainty in Deep Learning. NeurIPS'19
> >
> > $\newline$
> >
> > Q5.c. Incorrect y-axis value.
> >
> > Thanks for figuring out this typo, we have corrected it in the revised manuscript.
> >
> > $\newline$
> >
> > Q5.d. Code for CIFAR100 is not provided.
> >
> > We have included the code and some running log to the attachment. Feel free to check the details.

---

> > ### Comment · Reviewer_mbau · 2021-11-17
> > **Response**
> >
> > I will only focus on soundness issue in this comment.
> >
> > > reversibility based on Metropolis acceptance-rejection rules is always satisfied
> >
> > Compared to DEO, DEO* has enriched state space with a boolean value "whether swap has happened in current window" and an integer value "steps left in current window". Therefore, the old Metropolis acceptance criterion for DEO cannot apply to DEO*.
> >
> > > temperature $T_2 = \infty$ requires the learning rate $\eta=0$ We don't know if it is a well-defined process in the counterexample.
> >
> > 1. The identity kernel with $\eta=0$ is well-defined process, although the stationary distribution is not unique.
> > 2. My counter example proves that with simple local exploration kernel (which is easy to understand), the target distribution is no longer the stationary distribution.

---

> > > ### Author Response · Authors · 2021-11-17
> > > **Response to the soundness issue**
> > >
> > > Thanks for your response.
> > >
> > >
> > > Q1a (continued). **The old Metropolis acceptance criterion for DEO cannot apply to DEO**.
> > >
> > > Imagine we are running a Metropolis-Hasting (MH) sampler with $W$ iterations (or time) and the initial distribution $\mu_0$ (the empirical distribution at time 0) is the stationary distribution $\pi$. Denote by $\tau$ the random time for the first swap. It is clear that $\mu_t=\pi$ for **any** $t\in [0, \tau]$.
> > >
> > > **If $\tau\geq W$**, it is clear that $\tau_{W}=\pi$ and there is no swap.
> > >
> > > **If $\tau<W$**, mimicking the same style of DEO$_{\star}$, we terminate the MH sampler at time $\tau$ and there is one swap.
> > >
> > > It is safe to say $\mu_t=\mu_{\tau}$ for any $t\in [\tau, W]$ or $\mu_{W}=\pi$.
> > >
> > > Please correct me if the analogy is not appropriate (recall that Langevin diffusion is reversible and maps the invariant distribution $\pi$ to the same $\pi$).
> > >
> > >
> > > $\newline$
> > >
> > > Q1b (continued). Initialize two chains with target distributions $p_1$ and $p_2$, then run DEO* with one even window and one odd window. The **output distribution is no longer $p_1$ and $p_2$**.
> > >
> > > Putting aside if $\sqrt{2\eta \tau} \mathcal{N}(0, I)=\sqrt{0 \cdot \infty} \mathcal{N}(0, I)$ is well-defined, could you **detail the reasons** why the distribution is not $p_1$ and $p_2$ due to the use of a window $W>1$?
> > >
> > > Could you propose a counter-example that uses a regular temperature with $T\in (0, \infty)$? Because lots of algorithms yield a different property based on singular hyperparameters, like $T=\infty$.

---

> > > > ### Comment · Reviewer_mbau · 2021-11-17
> > > > **Response**
> > > >
> > > > > Please correct me if the analogy is not appropriate
> > > >
> > > > Unfortunately, authors are confused by their highly non-rigorous argument.
> > > >
> > > > As I said, **DEO\* has enriched state space**, therefore, distribution on parameter space itself is not enough to characterize this Markov chain.
> > > >
> > > > If I must explain how it doesn't work analogously, the window setting selectively suppress some future Metropolis acceptance steps, therefore latter Metropolis acceptance steps are not actually Metropolis acceptance step.
> > > >
> > > > > $\sqrt{2\eta \tau} \mathcal{N}(0, I)=\sqrt{0 \cdot \infty} \mathcal{N}(0, I)$
> > > >
> > > > This indeed seems buggy. In order to avoid that, I encourage authors just don't think about SGLD at all but think about an arbitrary local exploration kernel. It could be SGLD, random walk, Gibbs sampling, HMC or anything. The universe of local exploration kernels surely includes identity kernel, which means we change nothing.
> > > >
> > > > > a counter-example that uses a regular temperature with $T\in (0, \infty)$
> > > >
> > > > Of course. The temperature of infinity is just to facilitate thinking.
> > > > If we set $T_1=1, T_2=2$ with quadratic potential, things would be the same.
> > > > I will append a toy model to demonstrate that.
> > > > I also encourage authors to conduct experiment themselves to verify that.

---

> > > > > ### Comment · Reviewer_mbau · 2021-11-17
> > > > > **Toy Example**
> > > > >
> > > > > ```
> > > > > import numpy as np
> > > > >
> > > > > N = 100000
> > > > > std = 1.0
> > > > > a = 1.0 / 2 / (std ** 2)
> > > > >
> > > > > T1 = 1.0
> > > > > T2 = 2.0
> > > > >
> > > > > x1 = np.random.normal(0.0, std * np.sqrt(T1), N)
> > > > > x2 = np.random.normal(0.0, std * np.sqrt(T2), N)
> > > > > print(f"std:\nx1:{np.std(x1)}\nx2:{np.std(x2)}")
> > > > >
> > > > >
> > > > > def local_exploration_kernel(x, T):
> > > > >     return x
> > > > >
> > > > >
> > > > > e1 = a * x1 ** 2
> > > > > e2 = a * x2 ** 2
> > > > >
> > > > > W = 20
> > > > > already_swapped = np.zeros(N, dtype=bool)
> > > > > for _ in range(W):
> > > > >     x1 = local_exploration_kernel(x1, T1)
> > > > >     x2 = local_exploration_kernel(x2, T2)
> > > > >
> > > > >     x1_remain, x2_remain = x1[~already_swapped], x2[~already_swapped]
> > > > >
> > > > >     e1_remain = a * x1_remain ** 2
> > > > >     e2_remain = a * x2_remain ** 2
> > > > >     alpha_remain = np.exp(
> > > > >         np.minimum((1.0 / T1 - 1.0 / T2) * (e1_remain - e2_remain), 0)
> > > > >     )
> > > > >     swap = np.random.rand(*x1_remain.shape) < alpha_remain
> > > > >     x1_remain[swap], x2_remain[swap] = x2_remain[swap], x1_remain[swap]
> > > > >
> > > > >     x1[~already_swapped], x2[~already_swapped] = x1_remain, x2_remain
> > > > >     already_swapped[~already_swapped] |= swap
> > > > > print(f"std:\nx1:{np.std(x1)}\nx2:{np.std(x2)}")
> > > > > ```
> > > > > One run on my computer gives
> > > > > ```
> > > > > std:
> > > > > x1:1.0000463025248283
> > > > > x2:1.4135626071519831
> > > > > std:
> > > > > x1:1.3585688085074705
> > > > > x2:1.0735610562099602
> > > > > ```
> > > > > The std of x1 is no longer around 1, indicating a shift in distribution.

---

> > > > > > ### Author Response · Authors · 2021-11-23
> > > > > > **The stopping time leads to a bias that decreases in big data problems**
> > > > > >
> > > > > > We would like to express our sincere gratitude to *reviewer mbau*, who pointed out the important fact that stopping time, characterized by the event of the first swap, may affect the distribution, although it only affects the bias analysis in section C.2 and doesn't affect the main body of the paper. **Fortunately, the bias can be controlled and becomes much smaller in big data problems**. To help analyze the bias, we introduce an auxiliary scheme $\widehat{\text{DEO}_{\star}}$,
> > > > > >
> > > > > > which resembles the $\text{DEO}_{\star}$ scheme except that there are no restrictions on conducting at most one swap.
> > > > > >
> > > > > > $\newline$
> > > > > >
> > > > > > Note that $\text{DEO}_{\star}$ freezes at most $W-1$ attempts of swaps in each neighboring chain in each window,
> > > > > >
> > > > > > while $\widehat{\text{DEO}_{\star}}$ keeps all of these chances open.
> > > > > >
> > > > > > Recall that the bias-corrected swap rate follows $\widetilde S=O\big(S e^{-O(\sigma^2)}\big)$ by invoking Eq (16).  Following a similar technique in **analyzing the bias through the difference of acceptance rates** in Lemma 2 or [2], we can show that the bias in each iteration is at most $\widetilde S -\underbrace{\mathbb{P}(\text{Acceptance rate given frozen swaps})}_{=0}=O(e^{-O(\sigma^2)})$ and still adapts to our Theorem 2. As empirically validated in section 5.1 of [1], the variance can be in the order of millions in CIFAR datasets.  In other words, the $\textcolor{red}{\text{restrictions on at most one swap only lead to a mild bias in big data problems}}$ and is less of a concern compared to the local trap problems. Empirically, we have verified the relation between the bias and the variance of noisy energy estimators in a simulated example in https://pasteboard.co/SUn3sEa0oozb.png. The code is presented in the Bias study folder in the supplementary file and we have also modified the submission in section C.2 accordingly. We see that given the exact energy estimator, there exists a bias given extended windows (such as $W=2$ or $5$), however, $\textcolor{red}{\text{as the variance increases, we see a trend that the bias decreases significantly}}$.
> > > > > >
> > > > > > We note that such a swap strategy for multiple chain parallel tempering (PT) is desired, although not free of bias, because running the standard gradient Langevin dynamics or two-chain PT with the equivalent computational budget may not lead to the desired performance in more complex experiments due to the local-trap and meta-stability issues.
> > > > > >
> > > > > > The biggest advantage of this algorithm is the $\textcolor{red}{\text{user-friendly}}$ feature in incorporating the non-reversibility into parallel tempering for big data problems. Interestingly, our algorithm suggests that SG-MCMC with cyclic learning rates performs similar to a sub-chain of non-reversibility parallel tempering in big data, while our algorithm has more theoretical support and guidance on the accelerations, for example [3].
> > > > > >
> > > > > > We thank the reviewer again for your critical suggestions, interests, and time investment. Hope that we have addressed your concerns. We would like to respond to any further concerns that you may have to improve the manuscript.
> > > > > >
> > > > > > **Remark: section C.2 has been modified by including one more bias study. We also provide experiments on the validation of the relation between the bias of the scheme and the variance of noisy energy estimators (code is also provided)**.
> > > > > >
> > > > > >
> > > > > > [1] Non-convex Learning via Replica Exchange Stochastic Gradient MCMC. ICML'20.
> > > > > >
> > > > > > [2] Austerity in MCMC Land: Cutting the MetropolisHastings Budget. ICML'14.
> > > > > >
> > > > > > [3] Spectral Gap of Replica Exchange Langevin Diffusion on Mixture Distributions. arXiv:2006.16193v2. 2020.

---

> > > > > > > ### Comment · Reviewer_mbau · 2021-11-23
> > > > > > > **Response**
> > > > > > >
> > > > > > > I thank the author for our consensus on existence of this bias.
> > > > > > >
> > > > > > > > although it only affects the bias analysis in section C.2 and doesn't affect the main body of the paper
> > > > > > >
> > > > > > > I strongly encourage authors to highlight that "DEO* with noiseless energy function doesn't converge to target distribution" in the main paper.
> > > > > > >
> > > > > > > The authors proposed a new claim on scale of this bias.
> > > > > > >
> > > > > > > > Following a similar technique in analyzing the bias through the difference of acceptance rates in Lemma 2 or [2], we can show that the bias in each iteration is at most ... $O(e^{-O(\sigma^2)})$
> > > > > > >
> > > > > > > I think this claim is too vague and study in section C.2 seems not sufficient to support the claim.
> > > > > > > I am willing to see more rigorous analysis.
> > > > > > >
> > > > > > > I also have questions on the detail of this claim:
> > > > > > > * What's the window size dependence?
> > > > > > > * What's the ratio of this bias and swap rate? Could this bias diminish when compared to the swap rate?
> > > > > > > * Can we control the bias with noiseless energy function, i.e. $\sigma=0$?
> > > > > > >
> > > > > > > > The biggest advantage of this algorithm is the user-friendly feature
> > > > > > >
> > > > > > > I agree that the algorithm is compatible with stochastic gradient and stochastic energy functions.
> > > > > > > However, I hope that before introducing an algorithm into noisy big data setting, we can have a moderate understanding of how the algorithm behaves in simpler situations.

---

> > > > > > > > ### Author Response · Authors · 2021-11-23
> > > > > > > > **Response**
> > > > > > > >
> > > > > > > > Thanks for the valuable suggestions.
> > > > > > > >
> > > > > > > > We have emphasized such a statement in the revision.
> > > > > > > >
> > > > > > > > Note that the motivation we propose the generalized DEO scheme is that given correction to account for the noisy energy estimators with limited chains, the **theoretically correct swap rate** is too small and often yields no swaps (accelerations) at all in a finite time. To propose more accelerations, we have to sacrifice some accuracy sometimes to balance between acceleration and accuracy via tuning the target swap rate $\mathbb{S}$, which naturally **leads to the optimal window size $W_{\star}$** (window dependence) to achieve the optimal round trip time. Since the bias-corrected swap rate itself is quite small, the bias caused by intensionally freezing some swaps is rather negligible.
> > > > > > > >
> > > > > > > > Regarding the control of bias with noiseless energy function, we acknowledge it is an interesting direction and will leave it as future work. Nevertheless, given the exact energy estimator, the generalized DEO scheme may not be required anymore because the acceptance rate may become large enough given many chains to recover the standard DEO scheme without bias and more chains are always preferred in those cases as suggested by [1].
> > > > > > > >
> > > > > > > > In a nutshell, the noisy energy estimator in big data is a crucial reason for us to study the generalized DEO scheme. Although it induces a mild bias, it is less of a concern compared to acceleration requirements in large-scale non-convex big data problems.
> > > > > > > >
> > > > > > > > [1] Non-reversible parallel tempering: a scalable highly parallel MCMC scheme. 2021.

---

> ### Comment · Reviewer_mbau · 2021-11-17
> **Additional Question on Experiments**
>
> I just looked into code provided by authors, hoping to reuse their code to conduct numerical experiments to prove the soundness issue of their algorithm.
> However, during this process, I find strange behavior of the code that I cannot understand.
>
> **It seems the code cannot sample from a Gaussian distribution.**
>
> For the `Code/Simulations/NR-PT_v2.ipynb` file, I made following modification.
> I changed a very complex energy function
> ```
> def mixture(x):
>     energy = 0.2 * (x[0]**2 + x[1]**2) - 2 * (cos(2.0*pi*x[0]) + cos(2.0*pi*x[1]))
>     regularizer = ((x[0]**2 + x[1]**2) > 20) * ((x[0]**2 + x[1]**2) - 20)
>     return energy + regularizer
> ```
> into a quadratic function
> ```
> def mixture(x):
>     energy = (x[0]**2 + x[1]**2)*1
>     return energy
> ```
> therefore the result distribution should be Gaussian.
>
> After re-run the notebook with above modification, the result (https://pasteboard.co/fH71dHY0ljmf.jpg) is not satisfactory.
> Only ensemble SGLD gives similar result.
> Both DEO and DEO* don't give right distribution, indicating bugs in the implementation.
>
> Could the author explain why their code cannot be used to sample Gaussian distribution?

---

> > ### Author Response · Authors · 2021-11-17
> > **Tuning learning rate is crucial in the SGD-based uncertainty estimation**
> >
> > **Simply setting lr_low = 0.03 instead of lr_low = 0.003 can handle this issue, as shown in https://pasteboard.co/KQtvehmqk9YQ.png**.
> >
> > Our algorithm proposes a **user-friendly fashion** to conduct uncertainty approximation **solely by tuning learning rates $\eta$ and the target swap rate $\mathbb{S}$**, where $\eta^{(1)}$ controls the uncertainty approximation of the target distribution, $\eta^{(P)}$ takes care of the exploration in non-convex settings, and $\mathbb{S}$ balances between acceleration and accuracy.
> >
> > Admittedly, the approximation error may be larger than SGLD ensemble in convex scenarios due to the deterministic swap condition. We suggest the users apply this method in **non-convex** uncertainty estimation tasks in deep learning.

---

> > > ### Comment · Reviewer_mbau · 2021-11-17
> > > **Response**
> > >
> > > SGLD in the single chain case has been extensively studied. Previous works all show a negative relationship between step size $\eta$ and difference between stationary distribution and target distribution, i.e. the smaller step size is, the smaller error between stationary distribution and target distribution is. At infinitesimal step size limit, the SGLD could converge to true distribution.
> > >
> > > However, authors last response shows that in order to correctly sample from a Gaussian distribution, they need to choose a large step size for SGLD.
> > >
> > > What's the reason behind this novel observation? If we don't tune step size for exploitation chain, instead, at infinitesimal step size limit, will DEO* still converge to the correct Gaussian distribution?
> > >
> > > Even with large step size, output distribution from DEO and DEO* seems still worse than ensemble SGLD, is it normal?

---

> > > > ### Author Response · Authors · 2021-11-17
> > > > **Why the SGD-based parallel tempering performs worse than SGLD in convex scenarios.**
> > > >
> > > > The SGD-based parallel tempering includes two approximations (one is to convert SGLD to SGD, another is to convert Metropolis Hasting to Deterministic swap condition) to accelerate the simulation of multi-modal distributions in a user-friendly nature, as we stated in Theorem 2 and page 23.
> > > >
> > > > This is why the SGD-based parallel tempering performs worse than SGLD in **convex scenarios, in which we are not interested**. Tuning learning rates to achieve the best approximation is standard for the SGD-based samplers, as shown in [1]. It is task-dependent, not necessarily a large one.
> > > >
> > > > [1] Stochastic Gradient Descent as Approximate Bayesian Inference. JMLR. 2017.

---

> > > > > ### Comment · Reviewer_mbau · 2021-11-17
> > > > > **My question is specific to exploitation chain**
> > > > >
> > > > > I understand that for exploration chain, SGLD is approximated by SGD, and I agree that step size need to be large enough to introduce noise into the exploration chain.
> > > > > However, it seems exploitation chain is still SGLD, so my following question is specific to exploitation chain:
> > > > > > If we don't tune step size for exploitation chain, instead, at infinitesimal step size limit, will DEO* still converge to the correct Gaussian distribution?
> > > > >
> > > > > The authors, in previous comment, propose to tune `lr_low`. If I understand correctly, `lr_low` is directly used as step size for exploitation chain. Why the step size of exploitation chain need to be large?

---

> > > > > > ### Author Response · Authors · 2021-11-18
> > > > > > **Why the step size of exploitation chain need to be large?**
> > > > > >
> > > > > > Although SGLD is used in the simulation example, we still have **15 exploration chains based on SGD** and they may affect the overall approximation based on the deterministic swap condition.
> > > > > >
> > > > > > This is a reason why here the SGLD chain performs differently than usual. We believe replacing SGLD with SGD based on fine-tuned step sizes will produce similar results.

---

> > > > > > > ### Comment · Reviewer_mbau · 2021-11-18
> > > > > > > **Response**
> > > > > > >
> > > > > > > I agree that the exploration chain may affect the overall approximation in exploitation chain.
> > > > > > >
> > > > > > > I also agree that one reason for this undesirable impact is deterministic swap condition, as it is just an approximation of Metropolis acceptance step.
> > > > > > >
> > > > > > > I believe that window setting introduced in DEO* should be another reason. If the author are not convinced, we can discuss in the other thread that is specific to this issue.
> > > > > > >
> > > > > > > Overall, I and the authors all agree that the exploration chains could affect the stationary distribution in exploitation chain, instead of just accelerate the mixing.
> > > > > > > I currently also accept authors claim that the only reason that their implementation could not converge to Gaussian distribution with small step size is because of above impact.
> > > > > > >
> > > > > > > Now I just wish to emphasize that this impact deviates from the ideal parallel tempering, and should be reduced or at least controlled.
> > > > > > > I hope authors could highlight this phenomenon and pay more effort controlling this distribution error in their future revisions.

---

### Official Review · Reviewer_ofJx · 2021-11-04

**Correctness:** 3
**Technical Novelty And Significance:** 3
**Empirical Novelty And Significance:** 3
**Recommendation:** 5
**Confidence:** 3

**Main Review:**

I thank the authors for their article, which addresses the important topic of choice of swap schemes in parallel tempering. They build upon the current deterministic even-odd (DEO) swap scheme and develop modifications and generalisations which have more favourable theoretical and empirical performance.

Originality: The authors propose a generalisation of the existing DEO swap scheme. The details of their theoretical analysis and their methodological contributions are original and significant. Their investigation of stochastic approximations for big data, optimal number of chains, analysis of round trip time are detailed and through.

Clarity: The article is clearly written, even for readers who are less familiar with the swap schemes for parallel tempering.

I have minor comments about the paper. They are:
1) The approximation analysis theorem (theorem 2) assumes uniform ergodicity, which is stronger than “geometric ergodicity” as currently stated. Secondly, note that the bound given for theorem 2 is vacuous if (1-\rho)<\Delta. Some discussion of this point (that we require small \delta and \rho much less than 1 for the given upper bound \Delta/(1-\rho) to be useful) would be helpful.

2) There should be some discussion about the choice of W (or target swap rate S) in practice, what algorithms/ heuristics are recommended when choosing the parameter W, and what additional computational cost this incurs. For example, at the moment the optimal window size is given as a function of the number of chains and the target swap rate- but how is the target swap rate chosen? I could be misunderstanding something here.

3) If possible, the authors could present a version of Cor 1 which does not assume that the optimal W is chosen, but a generic W. Alternatively, some discussion about the importance of the choice of W should be included.

**Summary Of The Paper:**

The manuscript considers parallel tempering for sampling from multi-modal distributions. The choice of the swap scheme can great impact the performance of parallel tempering algorithms. The authors propose a modification of the existing deterministic even-odd (DEO) swap scheme. Theoretical results are established which show the proposed scheme improves communication cost from O(P^2) to O(P*logP) for P chains. Simulation studies are performed on a stylised multimodal distribution and on CIFAR100 datasets showing empirical improvements.

**Summary Of The Review:**

This manuscript proposes new swap schemes for parallel tempering. The methodology is supported by theoretical analysis, detailed discussion of methodological details and simulations. It will be of interest to the ICLR community.


EDIT: Having read the review of "mbau" and the corresponding discussion, I agree with the reviewer that the DEO* scheme may not converge to the target distribution of interest. I also agree the current argument in Section C.2 needs to be made more rigorous. Based on this point, I am inclined to revise my initial review score for the paper.

---

> ### Author Response · Authors · 2021-11-17
> **We appreciate your insightful suggestions.**
>
> Q1. Uniform ergodicity and dependence of $\Delta_{\max}$ w.r.t. hyperparameters.
>
> Thanks for pointing out this issue. One solution to overcome the discrepancy is to limit the parameter space to a bounded domain to facilitate the approximation analysis based on the uniform ergodicity, although the empirical performance is not much affected.
>
> Regarding the dependence of $\Delta_{\max}$ w.r.t. the hyperparameters, unfortunately, we cannot guarantee $\Delta_{\max}<1-\rho$ in the current proof. Although Lemma 2 shows that a correction term in the deterministic swap condition exists to perfectly approximate the Metropolis rule for each $\partial U_p$, even a well-tuned correction term may lead to a crude approximation in the worst case (although with a rather small probability). As such, the best we can achieve is that by fine-tuning the correction (through the target swap rate) and learning rates through cross-validation, we may obtain one such $\Delta_{\max}$ that is smaller than $1-\rho$.
>
> We will include relevant discussions in the next revision.
>
> $\newline$
>
> Q2. Hyperparameter tuning of the target swap rate $\mathbb{S}$ and the window size $W$.
>
> In non-convex learning of big data problems, there is a trade-off between acceleration and accuracy.
>
> If we propose the exact correction term in Eq.(16), the corresponding **bias will be minimized**, however, the **acceleration will disappear and no swaps can be proposed** in practical big data problems. For the approximation tasks with low-accuracy demand, **sacrificing accuracy to gain more acceleration in the early period** is often preferred. As such, we suggest to tune $\mathbb{S}$ via cross-validation, as demonstrated in the second paragraph of section 5.1 and the second paragraph of section 5.2.
>
> For each candidate $\mathbb{S}$, the optimal window size $W$ will be fixed according to the formula.
>
>
> $\newline$
>
> Q3. Discussions about the importance of the choice of W in Corollary 1.
>
> Thanks for pointing out this issue. As a matter of fact, losing the optimality of W may only slightly decrease the optimal number of chains $P_{\star}$, although it is still very costly to run $P_{\star}$ chains without the optimality of W for big data problems. We will include such a remark in the next revision.

---

### Official Review · Reviewer_pLfL · 2021-11-07

**Correctness:** 4
**Technical Novelty And Significance:** 3
**Empirical Novelty And Significance:** Not applicable
**Recommendation:** 8
**Confidence:** 2

**Main Review:**

**Pros**
1. The contribution is very concrete
2. $O(P)$ improvement is significant
3. The proposed algorithm is based on a slight modification of an existing algorithm
4. The paper is very well written
5. The proof is quite straight words
6. Literature is well-reviewed in section 2.
**My concerns**
1.  Is possible to modify the rejection probability in DEO to get the same rounding time (for example use the rejection $O(r^{\log(p)})$?
2. I could not understand why the paper focuses on the gradient Langevin dynamics. Is it possible to use a different sampling algorithm?
3. The notion of optimality is rather confusing in the abstract. Is it possible to show that it is not possible to get a communication cost lower than $O(P\log(P))$? I think that the optimality is about the optimal window size for the adjusted DEO.
4. I wondered whether authors use the real SGD algorithm or they use gradient descent + a Gaussian noise?
5. The transition from the main result to applications for SGD can be more detailed and more clear. This application can be motivated more and explain why this particular case of parallel tempering is studied in the paper. The difference between SGD and Euler's discretization of Langevin dynamics can be explained in more detail.
6. I recommend presenting experimental validations for Thm. 1 after this Thm. before talking about SGD.

**Summary Of The Paper:**

The paper studies the communicational complexity of parallel tempering. One example of parallel tempering is when one uses Langevin dynamics with different temperatures to balance exploit and exploration for sampling from a multi-modal distribution. The balance is made by swapping running particles with different temperatures according to a particular schedule. The round trip time (RTT) is a computational complexity metric for measure for the swapping schedule.  It is known that the best-known algorithm suffers from $O(P^2)$ RTT for $P$ parallel particles. By adjusting this algorithm, the authors improve the RTT to $O(P\log(P))$. Then, they demonstrate the application of nonconvex with stochastic gradient descent.

**Summary Of The Review:**

I liked the contribution and narrative of this paper, and my opinion about the result is rather positive.

---

> ### Author Response · Authors · 2021-11-17
> **We may not intentionally use a small rejection rate to get the same round trip time**
>
> We appreciate your valuable comments.
>
> Q1. Can we intentionally use a small rejection rate to get the same rounding time?
>
> No. As demonstrated by Fig.3 in this paper or Fig.2(d) in [1], manually setting a small rejection rate leads to larger approximation errors. Given the fixed highest and lowest temperatures with the **same bias budget, increasing the number of chains or devising an extended window to allow at most one swap seems to be the only way to achieve this target**.
>
> [1] Non-convex Learning via Replica Exchange Stochastic Gradient MCMC. ICML'20.
>
> $\newline$
>
> Q2. Is it possible to use a different sampling algorithm?
>
> Yes. We adopt gradient Langevin dynamics (GLD) sine it is standard and there is a rich literature. Variants of GLD, such as (stochastic gradient) HMC [1,2], underdamped Langevin [3], (approximated) Riemann manifold Langevin [4,5,6,7], can also be adopted to improve the performance.
>
> [1] MCMC using Hamiltonian dynamics. Handbook of Markov Chain Monte Carlo, 2010.
>
> [2] Stochastic Gradient Hamiltonian Monte Carlo. ICML'14
>
> [3] Underdamped Langevin MCMC: A Non-asymptotic Analysis. COLT'18
>
> [4] Riemann Manifold Langevin and Hamiltonian Monte Carlo methods. JRSSB, 2011.
>
> [5] Preconditioned Stochastic Gradient Langevin Dynamics for Deep Neural Networks. AAAI'16
>
> [6] Stochastic Gradient Riemannian Langevin Dynamics on the Probability Simplex. NIPS'13
>
> [7] Stochastic Quasi-Newton Langevin Monte Carlo. ICML'16.
>
> $\newline$
>
> Q3. Did the authors use the real SGD algorithm?
>
> To illustrate the performance of the proposed algorithm for a multi-modal distribution, gradient descent with Gaussian noise is adopted to mimic the behavior of SGD in section 5.1. Empirical evaluations of the **real (momentum) SGD** algorithms are conducted in section 5.2.
>
>
> $\newline$
>
> Q4. The notion of optimality is confusing.
>
> We acknowledge that the notion of optimality only applies to the generalized DEO scheme. Other swap schemes, such as APE with an expensive swap time of $O(P^3)$, may yield lower round trip time than O(P log P). We have adjusted our claim in the abstract to ``obtain an appealing communication cost $O(P\log P)$ based on the optimal window size.'' in the revised version.

---

> > ### Author Response · Authors · 2021-11-17
> > **Detail the transition and motivation of SGD as exploration kernels.**
> >
> > Note that efficient explorations do not only require a high temperature but also **prefer a large learning rate**.
> >
> > **Weakness of SGLD:**
> >
> > 1. SGLD is only theoretically appealing given a small learning rate; SGLD with a large learning rate leads to a large approximation error [1,2].
> >
> > 2. Tuning temperature itself is already quite costly even for one single chain. Parallel tempering with P additional temperature variables will require too many tuning efforts.
> >
> > **Strength of SGD** in sampling with **low-accuracy** demand:
> >
> > 1. Proven empirical successes in training almost every aspect of deep learning tasks;
> >
> > 2. Empirical experience in tuning learning rates can be incorporated into the design of PT. No need to tune temperatures.
> >
> > 3. Fixed learning rates yield attractive theoretical guarantees in sampling tasks with low-accuracy demand [3].
> >
> > $\newline$
> >
> > [1] User-friendly Guarantees for the Langevin Monte Carlo with Inaccurate Gradient. arXiv:1710.00095. 2017
> >
> > [2] Approximation Analysis of Stochastic Gradient Langevin Dynamics by using Fokker-Planck Equation and Ito Process. ICML'14.
> >
> > [3] Stochastic Gradient Descent as Approximate Bayesian Inference. JMLR, 2017.

---

### Author Response · Authors · 2021-11-17
**Update of the main paper and the appendix**

Thanks for your valuable comments and suggestions on the manuscript. We have slightly modified the abstract regarding the notion of optimality and included the experiment code including some running logs. We also fixed some typos.

---

### Decision · Program_Chairs · 2022-01-20

**Decision:**

Reject

**Comment:**

This paper proposes the algorithm which they call DEO*-SGD, which is a combination of the ideas of the generalized DEO scheme, denoted by DEO*, to facilitate exploration (Section 3.1), adoption of stochastic gradient descent (SGD) in the exploration chains (i.e., those chains except the one with the lowest temperature) (Section 4), and use of adaptive tuning of learning rates (Section 4.2). The proposal is applied experimentally in Section 5 to demonstrate superiority of the proposal over existing approaches.

The initial review scores of the four reviewers were one positive and three negatives. Most reviewers positively evaluated the proposal, including the proposal of DEO* and its theoretical analysis, as well as its empirical usefulness in deep learning for a computer-vision task. On the other hand, some reviewers showed concern about soundness of the proposal. Upon reading the reviews and the author responses, as well as the paper itself, I think that this paper lacks a clear statement on its objective.

* **What does "uncertainty approximation" mean?:** The paper title would imply that the objective of the proposal in this paper is for "uncertainty approximation," but I could not find any concrete description on what it exactly is.
* **Sampling versus optimization:** The methods of Langevin dynamics, or more generally Markov-chain Monte-Carlo methods, have been used for two distinct purposes: sampling and optimization. In any case fast relaxation towards equilibrium would be of practical importance. For sampling purposes it is also important to assure that the stationary distribution of the Markov chain corresponds to the target distribution (In Langevin dynamics the target distribution would be the canonical ensemble defined by the energy $U(\cdot)$ and the temperature $\tau$). For optimization purposes, however, the assurance of the stationary distribution to be equal to the target distribution would be less of concern. It seems that the authors' interest would be in optimization rather than in sampling, but it is not clearly stated.
* **Soundness issue:** As Reviewer mbau pointed out, DEO* does not have a guarantee of convergence to the target distribution. I thought that if the objective of this paper would be in optimization rather than in sampling, the existence of approximation already in DEO* might be thought of as a minor problem, as the proposal already has other approximations introduced in Section 4. The authors claim that this problem does not affect the main body of the paper, but I feel that it would affect the overall organization of the paper, as the current organization seems to presume that approximation only resides in the adoption of the SGD-based exploration kernels with deterministic swap. In any case, this problem has been acknowledged by the authors themselves, as well as Reviewer ofJx.

In particular, the detailed discussion between the authors and Reviewer mbau has been very fruitful in clarifying technical subtleties in this manuscript, including the soundness issue mentioned above. At the same time, it would imply that this paper still has room for improvement.

An additional point I would like to mention is that this paper is not really self-contained, in the sense that several key notions and quantities are not defined or only defined in the Supplementary Materials ($\tilde{U}$ is not explicitly defined at all, the terms "swap time" and "round trip time" are defined in Appendix A.5, $\sigma_p$ in Corollary 1 and Lemma 2 is defined in Appendix A.1).

All these weaknesses make me to think that another round of revision would be appropriate to properly judge the quality of this paper, whereas there is no such option within the review procedure of ICLR. I therefore cannot recommend acceptance of this paper at least in its current form.

Minor points (page and line numbers refer to the revised version):
- Abstract, line 5: "given sufficient many $P$ chains" would be better phrased as "given $P$ chains", as the big-O notation usually assumes the large-P asymptotic.
- In several places, there are periods after "Figure" and "Table", which are not needed.
- Page 3, line 32: In Lemma 2 there is apparently no such term found as "the second quadratic term". It should appear only after having assumed the equi-acceptance/rejection rates in equation (4), so that the sum becomes proportional to $P$.
- Theorem 1: "the maximal round trip time" should certainly be "the minimal round trip time". / is the ceiling function(. T -> , t)he round trip time
- Table 1: I did not really understand what "non-asymptotic" / "asymptotic" mean, as the big-O notation used here should by definition be asymptotic.
- Corollary 1: the optimal (number of) chains
- Page 4, line 34: The abbreviation SGLD is not defined in this paper.
- Page 4, line 36: similar(ly) to
- Equation (6): The sign of the last term should be "-".